| Open Peer Review | Bacteriophages | Methods and Protocols

# Phage DisCo: targeted discovery of bacteriophages by co-culture

Eleanor A. Rand,[1,2,3] Natalia Quinones-Olvera,[1,2,3] Kesther D. C. Jean,[1,2,3,4] Carmen Hernandez-Perez,[1,2,3,5] Siân V. Owen,[1,2,3,6] Michael Baym[1,2,3]

**ABSTRACT** Phages interact with many components of bacterial physiology from the surface to the cytoplasm. Although there are methods to determine the receptors and intracellular systems a specified phage interacts with retroactively, finding a phage that interacts with a chosen piece of bacterial physiology *a priori* is very challenging. Variation in phage plaque morphology does not to reliably distinguish distinct phages, and therefore many potentially redundant phages may need to be isolated, purified, and individually characterized to find phages of interest. Here, we present a method in which multiple bacterial strains are co-cultured on the same screening plate to add an extra dimension to plaque morphology data. In this method, phage discovery by co-culture (Phage DisCo), strains are isogenic except for fluorescent tags and one perturbation expected to impact phage infection. Differential plaquing on the strains is easily detectable by fluorescent signal and implies that the perturbation made to the surviving strain in a plaque prevents phage infection. We validate the Phage DisCo method by showing that characterized phages have the expected plaque morphology on Phage DisCo plates and demonstrate the power of Phage DisCo for multiple targeted discovery applications, from receptors to phage defense systems.

**IMPORTANCE** In this work, we describe a targeted phage discovery method that allows immediate isolation of phages with specific traits. Currently, to find a phage with specific properties, huge libraries of phages must be collected and screened retroactively. This assay, Phage Discovery by Co-culture (Phage DisCo), works by co-culture of host strains that are identical except for one perturbation that may interfere with phage infection and a unique fluorescent marker. These strains are co-cultured with an environmental sample of interest in traditional plaque assay format, making phage characteristics easily identifiable by fluorescent signal after imaging of the screening plate. We validate that Phage DisCo can identify phages with specific properties, even when these phages are rare in samples. This approach allows rapid exploration of the diversity within phage samples with vastly streamlined processes, and we anticipate it will be widely adopted within the phage discovery field.

**KEYWORDS** bacteriophages, phage receptor, phage defense, bacteriophage therapy

Address correspondence to Michael Baym, baym@hms.harvard.edu, or Siân V. Owen, sian.owen@health.ny.gov.

The authors declare no conflict of interest.

See the funding table on p. 13.

Bacteriophages (phages) are extremely abundant, diverse, and under-sampled in the environment (1). There are an estimated $10^{31}$ phages on Earth (2, 3), and, beyond their plentiful number, phages may represent some of the largest reservoirs of unexplored genetic diversity on the planet (4). Although this wealth of diversity can be sampled in almost any environment from soil to the ocean to the human gut (1), the distribution of phage diversity is not predicted to be uniform across environments (5). Instead, as has been recognized by ecologists since the 1940s (6), sequence-based analysis suggests there are huge differences in the abundance of different groups of phages, with a relatively large percentage of the phage population made up of highly abundant, similar phages while most diversity is found in low abundance, rare phages

(7). Current culture-based methods for phage discovery are intrinsically biased toward the abundant majority, and therefore increased phage discovery efforts in the absence of methodological innovation will inevitably lead to over-sampling of abundant phages, with only small and random glimpses at rare, phenotypically diverse phage populations.

Technologies have advanced to explore the characteristics of a given phage once it has been isolated in culture, for example, receptor determination (8, 9) or gene essentiality (10, 11), while the opposite problem, finding a phage that uses a specific receptor or with other specific characteristics from an environmental sample, remains a laborious task. Characteristics of interest may include interaction with or selection against a bacterial receptor protein of interest (12–15), finding a phage that interacts with a putative defense system (16), selecting a diverse cocktail of phages (17), or other similar quests. Methods to isolate phages with desired characteristics have not advanced significantly beyond the original plaque assay methodology (18). To find a phage of interest, a large panel of phages must be isolated on a single bacterial host strain, purified, replicated, and then screened retroactively on a range of bacterial hosts to identify candidates with various properties. The rarer the phage of interest, the more phages need to be isolated, processed, and screened.

Here, we describe a modification to the traditional plaque assay that allows for quick and efficient targeted isolation of environmental phages. We call this method phage discovery by co-culture (Phage DisCo). Phage DisCo works by culturing multiple bacterial strains in the same agar lawn and screening for differential plaquing via a fluorescence read-out. While we and others have previously demonstrated that this approach allows for the direct isolation of plasmid-dependent phages (19, 20), and fluorescence has been used to measure differential phage susceptibility between modified bacterial strains (21), here, we demonstrate its broader utility for general screening applications. With the ability to highlight differential lysis, we first validate the ability of Phage DiSco to distinguish between characterized phages T4, Bas10 (22), Bas37 (22), and U136B (12), which depend on OmpC, PqqU (previously YncD [23]), Tsx, and TolC, respectively. We also show that Phage DisCo can differentiate between variants of T4 which interact with restriction enzyme GmrSD (24, 25) as well as variants of temperate phage BTP1 which interact with phage defense system BstA (26). Second, we use Phage DisCo to isolate phages from the environment dependent on a number of *Escherichia coli* phage receptors, followed by the targeted isolation of phages that interact with specific phage defense systems. Given the generality of the method, we anticipate Phage DisCo to be readily adaptable to a wide range of targeted phage discovery applications.

## RESULTS AND DISCUSSION

### Phage discovery by co-culture

Because phage diversity in the environment is unlikely to be uniformly distributed, naive phage isolation will repeatedly recover the most common phages while only occasionally recovering rare phages. This is illustrated by the classically modeled rank abundance curve (Fig. 1A) showing that the most abundant phages make up a large proportion of the population while the bulk of the diversity is in the long tail of rare phages. Though phage plaque morphology offers a crude way to distinguish between different phages (27), it can be inconsistent and is associated with a small number of variables, making it an unreliable metric of diversity. In effect, current culture-based isolation methods are only able to sample the phage population, as opposed to sampling phage diversity.

To improve the efficiency of targeted phage discovery, we devised a co-culture screening approach which effectively adds an additional dimension to traditional plaque assay data. In its simplest form, the Phage DisCo assay uses two strains: a wild-type (WT) and a modified strain, each with their own fluorescent tag. The modified strain can be altered in any way that might interact with phage infection, for example, by knocking out or modifying a phage receptor or by introducing a known or putative phage defense system. The wild-type and modified strains are then plated together with

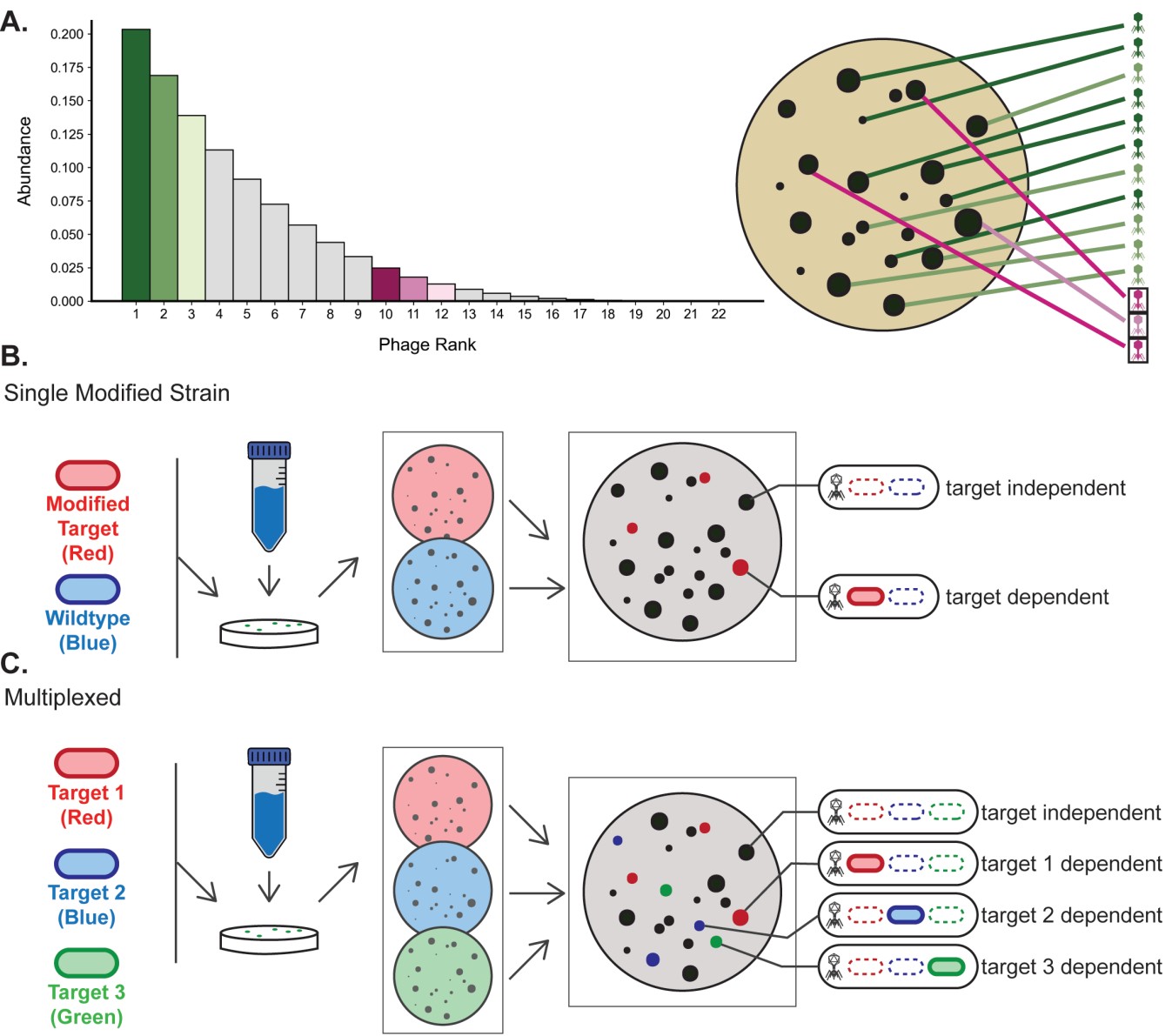

**FIG 1** Phage DisCo allows for rapid identification of phages with properties of interest. (A) Cartoon rank/abundance curve made by plotting decreasing integers raised to the fourth power to illustrate the expected distribution of diversity across phage taxa. Petri dish illustrates how this distribution of diversity would manifest on screening plates. If the pink phages were of interest, naïve sampling would likely uncover many green phages before a pink one. (B) Conceptual diagram of the most simple co-culture screening setup. In this case, a wild-type strain and a singly modified strain are each tagged with a unique fluorescent protein and plated together with an environmental sample. After imaging in the individual channels and creating a composite image, differential plating can be detected. (C) Conceptual diagram of a multiplexed co-culture system where three strains, each with a modification and a fluorescent tag, are plated together with an environmental sample. This allows for extra dimensionality on each screening plate.

an environmental sample containing unknown phages. After growth, the plate is imaged in both fluorescent channels, and plaques with differential fluorescence between the two strains are identified, most readily by a composite image (Fig. 1B). In dark plaques, where there is no detected fluorescence, both strains are being lysed by the phage, and thus, it is likely that the modification made in the second strain is not affecting phage infection. However, in colored plaques, where there is a fluorescent signal from the modified strain, only the wild-type strain is being lysed. In this case, it is likely that the alteration in the modified strain has interrupted the phage infection cycle (Fig. 1B). This technique enables rapid disregard of the plaques formed by off-target

phages and immediate identification of target phage plaques, greatly reducing laborious downstream characterization.

The DisCo method can be further multiplexed by adding more than two strains to the screening lawn. To increase screening throughput, three or more fluorescently tagged uniquely modified strains can be combined in a screening plate along with an environmental sample, limited by the number of distinguishable detection channels. As before, a dark plaque without fluorescent signal represents a phage that lyses all strains and thus likely does not interact with any of the modifications made to the bacterial strains. However, in this multiplexed design, there is not just one possible fluorescent plaque. Instead, there are multiple possibilities, most importantly single-color plaques in which the modification of the surviving tagged strain impedes phage infection (Fig. 1C). This approach allows the detection of multiple orthogonal phage traits in a single one-step screen. Note that if non-orthogonal traits are targeted (i.e., primary and secondary receptors), bi-color plaques can be detected. The extra information gained from co-culturing multiple strains has the potential to speed up targeted phage discovery by orders of magnitude while retaining many of the advantageous features of conventional methodology.

## Phage DisCo can recover characterized phages based on their known receptors

To validate that Phage DisCo works as expected, we first tested the co-culture setup with a pool of previously characterized phages (T4, U136B, Bas37, and Bas10) and *E. coli* knockout strains (modified from the Keio collection [28]) corresponding to each phage's receptor protein (Table S1). We imaged each channel using our custom fluorescent plate imager (Fig. S1; Table S2) and made a composite image. Each phage can be immediately identified by plaque color (Fig. S2). We note that there can be an apparent increase in fluorescence intensity within colored plaques relative to the mixed strain lawn due to the relative increase in single strain density when other strains are unable to grow, and this effect makes single-color plaques particularly easy to detect in composite images. Having established that Phage DisCo can distinguish between well-characterized phages, we proceeded to test our system with environmental samples containing unknown phages.

## Phage DisCo can efficiently isolate novel phages with specific receptor dependencies

To determine how our screen would work with unknown environmental phages, we used the same set of three strains used for the proof-of-concept experiment (Table S1) and added filtered samples of wastewater from the Boston metropolitan area (MA, USA). As with the characterized phages, after making composite images of all three fluorescent channels, we could immediately identify phages putatively dependent on each protein by choosing fully fluorescent plaques (Fig. 2A). To validate the receptor dependency of the wild phages identified in our DisCo screen, we picked a plaque of each color for further downstream processing. We note that without the extra layer of information provided by the fluorescent signal, many of these plaques have similar morphology, making them impossible to distinguish as unique. After purification and replication of each phage, we plated each on monoculture lawns of wild-type, knockout, and complemented knockout strains to validate receptor dependency (Fig. 2B, raw data in Fig. S3). As expected, each phage dropped below detection when plated on a receptor knockout strain, and plaquing was fully restored by plasmid complementation, validating our DisCo approach for receptor-guided phage discovery. Although some plaques have fluorescent halos, we do not expect the corresponding phages to have the full protein dependence as in the fully fluorescent plaques (Fig. S4).

To quantify the improved efficiency of using Phage DisCo to identify target-specific phages compared to traditional methods, we recorded the frequency of positive plaques for each receptor target. Of 746 total plaques screened across five replicates, 11 were

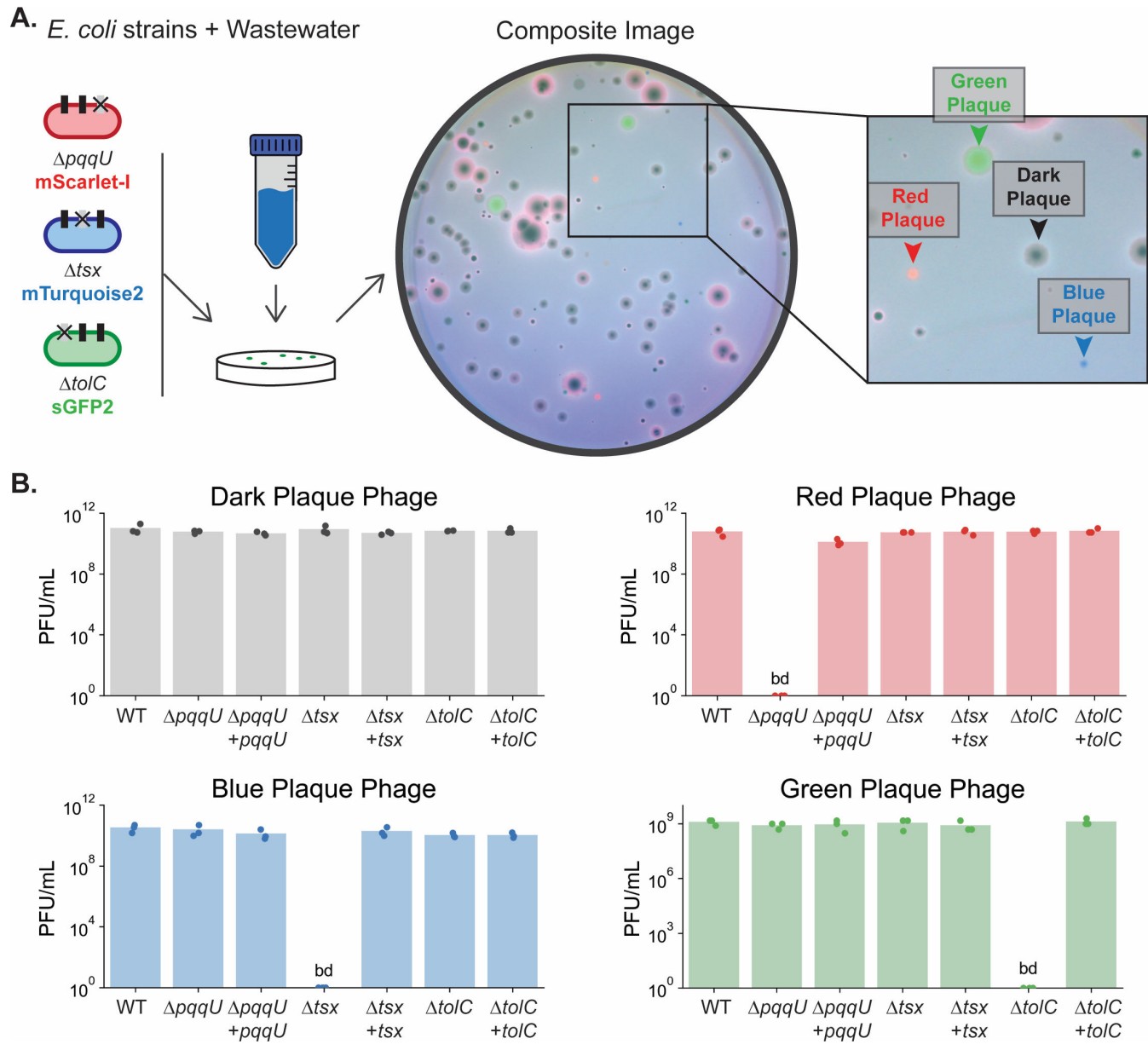

**FIG 2** Phage DisCo can discover receptor-specific phages. (A) Screening plate with three *E. coli* knockout strains and a wastewater sample. Arrows point to the plaques selected for analysis. (B) Each phage was purified, replicated, and three biological replicates were plated for plaque forming units per milliliter (PFU/mL) on seven monoculture bacterial lawns. The lawns were wild type (WT), each of the three knockout strains, and each of the three knockout strains plus plasmid complementation. Counts below detection are noted as bd.

TolC-dependent, 46 were Tsx-dependent, and 11 were PqqU-dependent. Given that picking plaques at random to find one dependent on a receptor of interest would follow a negative binomial distribution, it would be necessary to screen on average 67, 15, and 67 plaques before finding a TolC, Tsx, and PqqU phage, respectively, without the extra information provided by Phage DisCo. This result highlights that isolation efficiency using Phage DisCo compared to traditional techniques increases when phages of interest are rare. If a phage of interest is expected to make up a large proportion of the sample, Phage DisCo may not significantly improve isolation efficiency. However, when phages of interest are rare in an environment, Phage DisCo screens can increase the efficiency of isolation and screening depth substantially as compared to traditional techniques.

## Discovery of phages interacting with defense systems

Though receptor specificity dictates the initial outcome of phage infection, it is increasingly recognized that many other bacterial factors can affect the outcome of phage infection (29). We used Phage DisCo to discover phages that interact with phage defense systems. We used two isogenic strains per plate, where each has a fluorescent tag and one strain is expressing a known phage defense system. If a phage is sensitive to the defense system in question, the defense-negative strain will be fully lysed while the defense-positive strain will continue to grow and emit fluorescent signals, leading to detectable colored plaques. To test this screening method, we conducted a positive control using the phage defense system GmrSD, a modification-dependent restriction enzyme which cleaves DNA at the 5hmC modification found commonly in T-even phages (24) (Table S3). Phage T4 is natively insensitive to GmrSD because of the anti-defense internal protein IPI, which inhibits the effect of GmrSD (25). We tested whether phage DisCo could distinguish between GmrSD-sensitive and -resistant phages (T4 ΔIPI and T4 WT respectively).

As anticipated, T4 ΔIPI and T4 WT plated differentially on the co-cultured lawns (Fig. 3A and B), indicating that Phage DisCo plates can identify phages that interact with a specified phage defense system. No fluorescent signal was detected in plaques made by phage T4, but red fluorescent signal was visible where cells expressing GmrSD had resisted infection by phage T4 ΔIPI. In contrast to the phage DisCo screens for receptor-dependent phages, these plaques were associated with a unique fluorescent signal that was intense around the periphery of the plaque but depleted in the center, creating a halo effect. This effect is likely to be caused by the spatiotemporal dynamics of phage concentration across the plaque, and we speculate that the GmrSD defense system may be overwhelmed at high concentrations of phage, as has been previously observed (30). Having established the utility of the assay to detect GmrSD-sensitive and resistant phages, we next used the same assay to screen for wild phages in wastewater samples. Consistently, we were able to rapidly identify plaques with the same fluorescent halo morphology, and we confirmed that the identified phages were sensitive to GmrSD by purifying and plating them on monoculture lawns (Fig. 3C and D). Our isolation of wild GmrSD-sensitive phages demonstrates the potential for Phage Disco to identify phages that are interacting with restriction modification-based defense systems, which is an example of a defense system that presumably saves the life of the infected cell. However, recently, it has become clear that a large fraction of all known phage defense systems operate via abortive infection, wherein the infected cell dies before progeny phages can be released (31, 32). We reasoned that such mechanisms may be difficult to incorporate into the phage DisCo screening method because successful phage replication in the defense-negative strain would kill the co-cultured defense-positive strain by abortive infection, potentially preventing differential plaquing.

To test if the Phage DisCo screening approach would work for a known abortive phage defense system, we tested a recently characterized cyclic oligonucleotide-based antiphage signaling system (CBASS). Following infection with phages, CBASSs cause rapid cell death via activation of toxic effector proteins (33). To test if phage DisCo was able to detect CBASS-sensitive phages, we used a CBASS capable of defending against a broad range of phages (34). Despite the potential loss of signal from abortive infection, many putative CBASS-sensitive phages were detected in wastewater samples (Fig. S5). Two positive and two negative plaques were purified and screened in monoculture to validate the DisCo screen. Both putative CBASS-sensitive phages showed reduced efficiency of plating on strains containing the defense system in monoculture, validating the screen. However, we note that one of the putatively defense-resistant control phages (Dark Plaque Phage 1$^{CBASS}$) also showed lower plating efficiency on the CBASS-containing strain as compared to the wild-type strain, implying false negatives may occur with some defense systems (Fig. 3E). Our data do not suggest that the false negative was caused by phage stock contamination, as this would not explain a decrease in expected lysis levels. Similarly, data do not show that overexpression of CBASS from a plasmid contributed

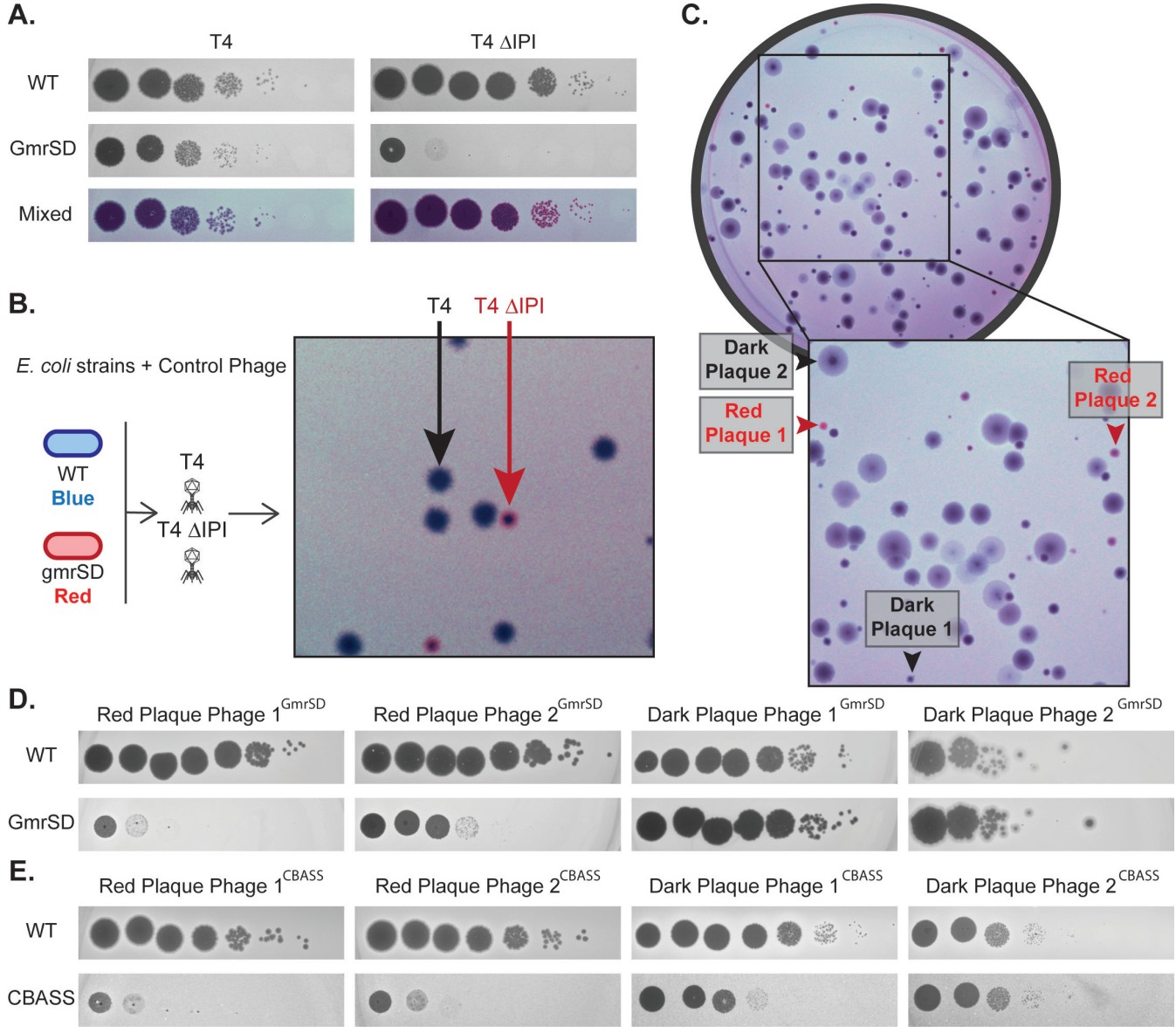

**FIG 3** Phage DisCo can discover phages that interact with specific phage defense systems. (A) Monoculture and mixed lawns with control phages to indicate expected plaque morphology when screening for phage defense system-sensitive phages. (B) Plaque morphology of controls on a simulated Phage DisCo screening plate with known phages. (C) Wastewater screening plate with the same strains shown in panel A. Two hits are labeled as Red Plaque 1 and 2, and two non-hits are labeled Dark Plaque 1 and 2. (D) Each of the phages in the plaques highlighted in panel B after purification and replication on monoculture WT and GmrSD containing lawns. (E) Phages picked from a similar screen for phages that interact with a cyclic oligonucleotide-based antiphage signaling system (CBASS). The screening plate for this assay is shown in Fig. S5.

to the false negative (Fig. S6). While we were able to detect phages sensitive to CBASS with our phage DisCo assay, sensitivity may be reduced for abortive infection systems, resulting in decreased ability to detect phages sensitive to this type of defense system.

As a final test of Phage DisCo, we screened for phages that interact with the abortive phage defense system BstA in *Salmonella enterica* (26). While we were able to validate that Phage DisCo can determine the difference between phages sensitive to BstA, including temperate phage BTP1 (Fig. S7), no BstA-sensitive environmental phages were detected in our screens. This suggests that BstA-sensitive phages may be rare in the environmental samples screened, though we cannot rule out the possibility of false negatives, as observed for the CBASS.

## Conclusion

In this paper, we have expanded upon the widely used traditional plaque assay for phage screening and isolation by co-culturing multiple fluorescently tagged bacterial host strains in a single lawn to create an efficient, targeted assay. We validated this assay by testing previously characterized phages and showed that each phage was distinguishable by its known receptor. Our experiments also showed that we could identify phages from environmental samples with desired characteristics in a single screening step, which drastically cuts down on the steps and time required by current techniques. We also validated that the method can distinguish between control phages that do and do not interact with a given defense system, and in two cases found uncharacterized environmental phages that interact with these defense systems. Based on work presented here and previously published work, we conclude that Phage DisCo can substantially reduce time and labor needed for targeted discovery of phages from the environment. However, we note that once phages of interest have been isolated, they should be purified, replicated, and characterized fully before further use.

We can see possible applications for Phage DisCo across multiple fields. Targeted screening methods can help survey ecological relationships across environments and within the same environment over time. Understanding phages within an environment could help us understand some of the selective pressures on bacteria, for example, what defense systems may be beneficial or redundant in contemporaneous bacterial populations. Screening for differential plaquing across strains can also give insights into interactions between host genes such as receptors and phage infections. This can be utilized to quickly compile diverse phage cocktails for phage therapy, as using phages with diverse receptors within a phage cocktail is thought to avoid easy pathways to resistance (35). Similarly, there have been suggestions to ensure that all phages in a cocktail are not all sensitive to the same defense system (36). Phage DisCo can quickly survey an environment for phages dependent on diverse receptors as well as interacting with diverse defense systems, as shown here by screening the same environmental sample on separate sets of indicator strains. Future studies could nest Phage DisCo screens by first isolating a pool of phages dependent on specific host protein(s) of interest and then screening that pool to determine a second trait such as interaction with specific defense system(s).

Here, we have added defense systems on plasmids to a lab strain *E. coli*, but modifying environmental isolates may help discover more relevant phages. For example, instead of heterologously expressing a defense system in K-12 *E. coli,* pairing wild-type bacterial isolates containing defense systems together with isogenic knockouts could allow more physiologically relevant screens, potentially increasing the positive hit rate. To illustrate this point, if the goal is to isolate a medically relevant phage to treat human infections, screening on a co-cultured isogenic pair of clinical bacterial isolates will likely lead to a more successful screen as compared to a pair of lab bacterial strains. Furthermore, the use of Phage DisCo to isolate defense system-sensitive phages has the potential to provide insights into the molecular mechanisms of these systems themselves. For instance, once isolated using Phage DisCo, mutants of defense system-sensitive phages could be evolved by passaging through bacterial strains that contain the corresponding defense system and studied to pinpoint how the phage and defense system interact (37).

So far, we have tested Phage DisCo with a maximum of three strains per screening plate, but we expect that with the correct imaging setup, this number could be increased. The maximum number of strains that could be successfully screened using Phage DisCo would be set by either the maximum number of strains that can coexist on a plate at high enough density to have detectable fluorescent plaques or the number of distinguishable fluorescent markers that each strain can be tagged with (whichever is limiting). Increasing the number of strains may require optimization of cell concentration in the lawn, as lower cell concentrations can lead to granularity of the fluorescence signal in *E. coli*. However, we found that fluorescent plaques are detectable with 100× fewer cells than optimal (Fig. S8). This suggests that plaquing could be observed with over 100

strains on a plate and still allow for fluorescent signal in plaques of interest. Currently, there are only around seven spectrally distinguishable fluorescent markers that can be used in bacteria (38). Therefore, if using all currently available fluorescent markers with an imaging setup capable of resolving each signal, Phage DisCo could be scaled to include up to seven strains per plate.

Although we anticipate Phage DisCo will be generally applicable to targeted phage discovery efforts across many bacterial study systems, there are limitations. To set up a Phage DisCo screen, genetic tractability is required to introduce fluorescent tags and genetic perturbations between the isogenic bacterial strains. Genetic approaches for manipulating non-model bacteria are always improving, and, in the meantime, there are tools to introduce fluorophores across many bacterial phyla via transposons (38) and across Gammaproteobacteria via conjugative plasmids (19) as well as tools for allelic exchange across Proteobacteria (39). Another limitation, in the case of multiplexed screens using multiple knockout strains on the same plate (Fig. 1C), is that we assume the proteins knocked out in each strain are independent of each other within the bacterial host and during phage infection. If this is not the case, and the proteins are in some way dependent on each other, a phage that depends on any one of them will fail to make a plaque at all, and false negatives can occur. Replacing one knockout strain with a wild-type strain can reduce such false negatives at a slight cost to throughput. Similarly, care must be taken when interpreting data from Phage DisCo assays biologically: phages frequently have multiple receptors, which often include genetically complex structures such as lipopolysaccharides (40–43). Phage DisCo only shows the impact of a single genetic perturbation on phage infection, and perturbation of genes associated with epistatic effects (e.g., genes in the LPS biosynthetic pathway) may be difficult to interpret in terms of phage biology. In this case, follow-up, for example using knockout, knockdown, or overexpression methods (8, 9, 11), would be needed to verify the full dependencies of any given phage.

Phage DisCo has the potential to rapidly rule out common phages in any given sample and highlight rare phages with desired characteristics, improving the efficiency of culture-based virus discovery. We believe this method will vastly improve phage prospecting and help lead to discoveries in phage research.

## MATERIALS AND METHODS

### Bacterial strains, plasmids, and phages

*E. coli* knockout strains (Δ*pqqU*, Δ*tsx*, and Δ*tolC*) were taken from the Keio collection (28), and the complementation plasmids were taken from the ASKA collection (44). Correct identity of the Keio strains was confirmed using PCR amplification of the knockout region followed by sequencing, and ASKA strains were confirmed by extraction with the NEB Monarch Plasmid Miniprep Kit followed by sequencing. The pEB2 plasmids were adapted from Addgene #104007 to change kanamycin resistance to chloramphenicol resistance, which is compatible with the Keio strains using overlap extension PCR cloning to construct the final plasmid (45, 46). The mScarlet-I fluorescent protein was switched for mTurquoise2 and sGFP2 using the same method. The pEB2 plasmids were pooled together and then transformed into the Keio strains by electroporation. Briefly, strains were grown in LB broth without salt (10 g/L tryptone, 5 g/L yeast extract) until an optical density ~0.5, washed in cold, sterile water, and electroporated at a resistance of 200 ohms, capacitance of 25 μFD, and 1.6 volts. Transformants were selected for on 2% LB agar plates + 20 μg/mL chloramphenicol. GmrSD was amplified out of MRSN781659 cloned into the low copy pSC plasmid backbone using NEBuilder HiFi DNA Assembly Master Mix. The pBR322 plasmids were created by modifying the NEB vector (catalog #N3033S) (Summary of the plasmids used and modifications made is in Table S4.) All strains were grown in LB Lennox broth (10 g/L tryptone, 5 g/L yeast extract, 5 g/L NaCl). ASKA plasmids were selected for with 20 μg/mL chloramphenicol. No selection was required for the pEB2 or pBR322 plasmids after cloning.

## Wastewater collection and processing

Primary effluent wastewater samples were collected from two separate sites near Boston, MA, USA (prohibited under sampling agreement from publishing exact coordinates). The wastewater originated from both commercial and residential areas and contained mostly liquid with minimal biomass. Samples were centrifuged at 4,000 × $g$ for 30 minutes, and the supernatant was filtered through a 0.22 µm vacuum filter. Filtered wastewater was stored at 4°C until use. Classical double-layer plaque assays were used to titer the total detectable phages in each sample by adding 10, 100, and 300 µL wastewater to 100 µL of saturated wild-type *E. coli* BW25113 culture. The cells and wastewater were combined with 3 mL of LB Lennox top agar (LB Lennox media + 0.5% agar) at 55°C and poured onto a solidified LB + 2% agar plate. Plates were incubated at 37°C overnight to enumerate phage titers. Optimal plates have ~150 plaques per 10 cm Petri dish; this is approximately the maximum number that can be screened per plate without frequent overlap between plaques.

## Phage DisCo plaque assay

For the receptor screen, colonies of each fluorescently tagged Keio strain were added to 2 mL of LB Lennox broth without selection and grown until the culture was saturated. A total of 100 µL of each culture was added to 100–300 µL of filtered wastewater sample. Volume of wastewater was optimized based on total detectable phage concentration by the titer method described above and varied slightly depending on wastewater sample date and location. To make each screening plate, 3 mL of LB Lennox top agar at 55°C was added to the cells and wastewater mixture and poured onto a 10 cm Petri dish with solidified LB + 2% agar incubated at 37°C overnight.

For the GmrSD screening plates, a similar protocol was followed with the addition of arabinose at a final concentration of 0.1% to induce production of the GmrSD protein. Likewise, arabinose was added to induce the production of the CBASS construct, and anhydrotetracycline was added at a final concentration of 500 ng/mL to induce the production of the BstA construct (26).

## Fluorescent imaging

After overnight growth, Phage DisCo plates were imaged in our custom fluorescent plate imager (diagram and parts list in Fig. S1 and Table S2, respectively). In short, the plates are illuminated by colored LEDs with excitation filters (EX), the emitted light passes through emission filters (EM), and images are taken on a Canon EOS camera. The red channel pairs 567 nm LED with 562 nm EX and 641/75 nm EM filters, the green channel pairs 490 nm–515 nm LED with 494 nm EX and 540/50 nm EM filters, and the blue channel pairs 448 nm LED with 438 nm EX and 483/31 nm EM filters (summary of which channels were used to detect which fluorescent signals in Table S4). Exposure times varied between 0.5 s and 8 s depending on the number of strains on the plate and the intensity of the fluorescent signal. Exposure times were selected such that there was little to no fluorescent signal within the plaques in each channel while maintaining bright signal on the rest of the plate. Camera settings were kept at an aperture of 5.6 and ISO of 200.

Once the individual channels were imaged, composite images were created in Adobe Photoshop. The green channel containing GmrSD and pBR322-mWatermelon was remapped to the blue channel to enhance visual contrast. When necessary, channels were linearly adjusted for contrast to increase signal to noise. No conclusions were made using fluorescence intensity. Hits are based solely on the presence or absence of fluorescent signal.

## Purifying phages from Phage DisCo screening plates

Once hits were identified in composite images, phages were picked from plaques into 200 µL of SM buffer (200 mM $NaCl_2$, 10 mM $MgSO_4$, 50 mM Tris-HCl, pH 7.5). The

resuspended plaque was then filtered using 0.22 µm Spin-X centrifuge tube filters. The filtrate was serially diluted in more SM buffer and then plated on monoculture lawns to confirm the predicted characteristics. These lawns were made by mixing 100 µL of cells and 3 mL of LB Lennox Top Agar at 55°C, with arabinose if required, and pouring the mixture onto a solidified 2% LB agar petri dish. A total of 2 µL of each dilution was spotted on top of the solidified cell and agar mixture. These spots were allowed to dry, and then the plates were incubated at 37°C overnight. Once the phenotype was confirmed, phages were replicated by adding 2 µL of phage to 2 mL saturated bacterial culture diluted 1:1,000 in LB broth. These new cultures were grown overnight at 37°C with aeration, then centrifuged at 16,900 × $g$ for 5 minutes and the supernatant passed through a 0.22 µm filter to remove bacterial cells. These high titer stocks were used to make another set of dilution plates following the same protocol for figures and stored at 4°C.

## Complementation assays

Complementation assays were done using the Keio background strain (*E. coli* BW25113), the Keio knockout strains, and Keio knockout strains plus the corresponding ASKA expression plasmids. To make the complemented knockout strains, the corresponding ASKA strains were grown in LB Lennox broth with 20 µg/mL chloramphenicol. Plasmids were extracted from the ASKA strains using the New England Biolabs Monarch Plasmid Miniprep Kit (catalog #T1110L) and transformed into Keio strains by electroporation as above. To test the plaquing ability of each phage, three sets of serial dilutions were made to provide biological replicates. The dilutions were arrayed in a 96-well plate, and lawns were made using each strain on rectangular Omnitrays (Nunc). Each lawn had 100 µL of cells and 5 mL of LB Lennox Top Agar at 55°C. Plates for the Δ*tsx*+*tsx* and Δ*pqqU*+*pqqU* strains were made using 20 µg/mL chloramphenicol in the 2% LB agar. A total of 2 µL of each of the dilutions was spotted onto each lawn using a Gilson Platemaster (96-well pipette). Spots were allowed to dry, and then plates were placed in plastic bags to prevent drying of the lawns during the overnight incubation at 37°C. Plates were then scanned using an Epson Perfection V850 Pro Scanner.

## Example protocol

### *Materials required*

- Bacterial strains
    - Fluorescently tagged either chromosomally or on a plasmid
        - Note: fluorescent tags must be constitutively expressed, and we found the extra signal provided by multicopy plasmids as compared to a single copy on the chromosome to be helpful
    - At least one strain with a genetic perturbation expected to impact phage infection
- Media and agar
    - Media for the chosen bacterial host
    - Petri dishes containing media + solidified 2% agar
    - Molten media + 0.5% agar (top agar)

### *Equipment required*

- Fluorescent imager
    - Capable of imaging full dishes at macroscopic level
    - Filters compatible with chosen fluorescent tags
    - Custom imagers can be built if a ready-made one is not available (47)

## Protocol

1. Inoculate one colony of each strain into individual aliquots of media and allow to grow until saturated. Time, temperature, aeration, etc. will depend on the bacterial host.
2. Once grown, mix equal volumes of each strain for each DisCo plate.
    - Note: volume needed to make a smooth lawn of bacteria will depend on the bacterial host. For lab *E. coli* strains used here, 100 µL of each strain was added to each plate.
3. Add filtered environmental sample to cell mixture.
    - Note: volume of environmental sample will depend on the concentration of phages. See the "Wastewater collection and processing" above for titer methods.
4. Add enough of the molten top agar to cover the screening dishes.
    - Note: for 10 cm petri dishes, 3 mL of top agar is recommended. For 128 × 86 mm rectangular dishes (Society for Biomolecular Screening [SBS]-sized plate), 5 mL of top agar is recommended.
5. Pour the cell, sample, and top agar mixture onto the solidified 2% agar dish, being careful not to add any bubbles.
6. Allow to set at room temp for ~2 minutes.
7. Incubate overnight at the correct temperature for optimal growth of the chosen bacterial strain.
    - Note: incubating inside an airtight plastic bag can help reduce drying for longer incubations and make images more even.
8. The next day, remove from the incubator and image each fluorescent channel individually.
9. Combine the channels into one image to detect differential plaquing.

When using lab *E. coli* strains, equal volumes of each strain were added to each screening plate without normalization for optical density or CFU. Even when strains grew at different rates, differential screening was detectable as a consequence of strains growing denser (and brighter) within partial plaques. This is particularly true if including a wild-type strain on a two-strain DisCo plate or if including three orthogonally perturbed strains on a DisCo plate. If using the Phage DisCo method under different conditions or with another bacterial host, it may become more important to control for these factors and others such as log phase growth. If lawns look patchy or grainy or if signals are hard to detect, it is recommended to grow cells to log phase (48) and then concentrate by centrifuging the liquid culture at 16,900 × *g* for 5 minutes, removing the supernatant, and resuspending in 10× less volume. After concentrating, continue with the method as written above (Fig. S8).

## ACKNOWLEDGMENTS

We thank Lucy McCully, Fernando Rossine, and Kepler Mears and the rest of the Baym lab for helpful discussions and materials. We thank Anurag Limdi for suggesting the name Phage DisCo. We thank Daniel Eaton and Johan Paulsson for making both the Keio and ASKA collections available to us, Alita Burmeister and Paul Turner for sharing a sample of phage U136B, Akos Nyerges and George Church for providing phage samples, and Peter Wiegele and Rebekah Silva at NEB for the modified T4ΔIPI phage. We thank Sam Hobbs, Desmond Richmond-Buccola, and Philip Kranzusch for useful discussions and providing resources. We thank the staff at the Multidrug-Resistant Organism Repository and Surveillance Network (MRSN) for providing strain MRSN 781659. The custom fluorescent imager was built using tools and with assistance from the Research Instrumentation Core at Harvard Medical School.

This work was supported by the NIGMS of the National Institutes of Health (R35GM133700 and R35GM156320), the David and Lucile Packard Foundation, the

Pew Charitable Trusts, Alfred P. Sloan Foundation, and NSF grant IOS-2331228. E.A.R. acknowledges support from the Systems, Synthetic, and Quantitative Biology PhD program training award (T32GM135014) and from NIH NIAID F31 (F31AI178993).

## AUTHOR AFFILIATIONS

[1]Department of Biomedical Informatics, Harvard Medical School, Boston, Massachusetts, USA

[2]Laboratory of Systems Pharmacology, Harvard Medical School, Boston, Massachusetts, USA

[3]Department of Microbiology, Harvard Medical School , Boston, Massachusetts, USA

[4]Roxbury Community College, Boston, Massachusetts, USA

[5]Summer Honors Undergraduate Research Program, Harvard Medical School, Boston, Massachusetts, USA

[6]Division of Genetics, Wadsworth Center, New York State Department of Health, Albany, New York, USA

## AUTHOR ORCIDs

Eleanor A. Rand  http://orcid.org/0000-0003-0775-2064
Siân V. Owen  http://orcid.org/0000-0001-5330-3177
Michael Baym  http://orcid.org/0000-0003-1303-5598

## FUNDING

| Funder | Grant(s) | Author(s) |
|---|---|---|
| National Institute of General Medical Sciences | R35GM133700, R35GM156320, T32GM135014 | Eleanor A. Rand |
| | | Natalia Quinones-Olvera |
| | | Kesther D. C. Jean |
| | | Carmen Hernandez-Perez |
| | | Siân V. Owen |
| | | Michael Baym |
| David and Lucile Packard Foundation | Packard Fellowship | Michael Baym |
| Pew Charitable Trusts | Pew Biomedical Scholarship | Michael Baym |
| National Science Foundation | IOS-2331228 | Eleanor A. Rand |
| | | Natalia Quinones-Olvera |
| | | Kesther D. C. Jean |
| | | Siân V. Owen |
| | | Michael Baym |
| National Institute of Allergy and Infectious Diseases | F31AI178993 | Eleanor A. Rand |

## DATA AVAILABILITY

All raw data is found in the supplemental material, and all plasmids used were sequence verified using long read sequencing.

## ADDITIONAL FILES

The following material is available online.

## Supplemental Material

**Supplemental figures and tables (mSystems01644-24-s0001.docx).** Figures S1 to S8; Tables S1 to S4.

## Open Peer Review

**PEER REVIEW HISTORY (review-history.pdf).** An accounting of the reviewer comments and feedback.

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
