## [Reviewer comments · mSystems]

Phage DisCo: targeted discovery of bacteriophages by co-culture

Eleanor Rand, Natalia Quinones-Olvera, Kesther Jean, Carmen Hernandez-Perez, Sian Owen, and Michael Baym

Corresponding Author(s): Michael Baym, Harvard Medical School

Review Timeline:

Submission Date:	December 4, 2024
Editorial Decision:	January 8, 2025
Revision Received:	February 19, 2025
Editorial Decision:	March 21, 2025
Revision Received:	April 9, 2025
Accepted:	April 30, 2025

Editor: Alejandro Reyes Munoz

Reviewer(s): The reviewers have opted to remain anonymous.

Transaction Report:

DOI: <https://doi.org/10.1128/msystems.01644-24>

Re: mSystems01644-24 (Phage DisCo: targeted discovery of bacteriophages by co-culture)

Dear Dr. Michael Baym:

The manuscript has been reviewed by two independent referees and both agree that although the manuscript has potential it does require some modifications before considering it for publication.

Revision Guidelines

Sincerely,
Alejandro Reyes Munoz
Editor
mSystems

Reviewer #2 (Comments for the Author):

Please see attached.

Manuscript title: *Phage DisCo: targeted discovery of bacteriophages by co-culture*

Authors: Eleanor A. Rand, Natalia Quinones-Olvera, Kesther D. C. Jean, Carmen Hernandez Perez, Siân V. Owen, Michael Baym

Overall comment:

The authors build on their prior published (Phage Disco, PMID: 36993299) method to differentially identify phages based on their host-receptor utility for infection, and with host-defense sensitivity based on plaque assays involving co-cultures of bacterial strains with each strain carrying a fluorescent tag. This paper is well written and provides clear figures and data to present Phage Discovery by Co-culture (or Phage DisCo) method fitting the journal scope. We need improved methods for sampling phage diversity we observe by genomics. So this paper definitely provides a new handle towards this goal. However, claiming this method will solve all of our challenges in isolating novel phages and help in sampling phage diversity from an environment is far-fetched. Authors agree that there are obvious limitations to the method (target host needs to be engineerable etc), but claim that this method is suitable for surveying “ecological relationships across environments” Line 278. We recommend authors to either address this by detailing how they would go about building ecological relationships across environments and sample phage diversity or tone down their claims.

Here we provide the following comments and hope these comments will help authors to improve their report.

1. Based on the engineering needed for this method to work, we believe this method is best limited to model strains such as *E coli*/Salmonella. Extending this to diverse environmental bacterial isolates is unrealistic because of the unavailability of genetic tools (plasmids, promoter-RBS parts, reporters, selection markers, gene deletion methods). There is also a limit on the number of bacterial cultures that can be mixed in the co-cultures for the multiplex mode. Another point that needs consideration is potential co-culture interactions (e.g. growth promoting and/or inhibitory activity of one strain on another) and how they impact which phages can be isolated for the different strains. Anyways, Isolating phages on *E coli* is not going to provide us a catalog of phage diversity in the environment. For example, using a traditional phage isolation method using a single bacterial strain this report (PMID: 34784345) isolated highly diverse coliphages.
2. We do not agree (for reasons stated in #1 above) that Phage Disco can “quickly survey an environment of both axes of diversity” line 287. The authors predominantly use relatively well characterized hosts and phages. However, for the vast majority of phages, not much is known about what host genes or genomic regions are important for phage infection (let alone, rare phages which the paper's method claims to be useful in isolating and characterizing). Furthermore, given the vast number of genes or genomic regions involved in phage infection, co-interactions, polarity that may be involved with the genes, etc. a significantly larger number of plaque assays/plates would need to be utilized to understand phage infection mechanisms using this method, potentially reducing throughput.
3. Even if the *Phage Disco* method makes it easy to identify phages that use a defined receptor, it is still necessary to isolate and sequence the genome of those phages to truly

know the phage diversity. In that case, we don't think traditional culture-based method for phage discovery will be longer than designing an engineered host (by deleting a gene and developing fluorescent tag plasmid that works in that target host) compatible with the Phage Disco method. By using a 96 well plate Viral gDNA extraction kits and based on the average numbers mentioned line 184, at least one of each plaques identified as TolC-, Tsx- or PqqU-dependent in the environmental sample assay, could be identified.

4. Given the fact that fluorescent markers need to be genetically expressed in the (model) strains, what is the maximum limit on the number of strains that can be co-cultured with marker, such that fluorescence detection can be used to unambiguously identify plaques and what host strains they are lysing based on different colors/mixed colors?
5. The 46 Tsx-dependent plaques out of 746 representing 6% of all the screened plaques doesn't seem rare, especially because the entire diversity of the 746 plaques is not known. Moreover, are the plaques with a red halo shown in Figure 2A PqqU-dependent plaques? In that case the number of these plaques is bigger than 11 making those plaques less rare. The authors should provide information concerning those plaques to make a claim of diversity.
6. We are intrigued by the false negative data on CBASS expt (line 241). We wonder if that may be because the CBASS system is expressed from a plasmid, and the overexpression of CBASS is causing this false negative effect. Authors should add clarification to this point in the last section of discussion (line 300-315).

Minor comments

- Figure 1A: The petri dish illustration should be explained in the legend. I believe authors want to show that phage plaque morphology can be an inconsistent metric for differentiating phages, but it is not clear.
- Line 64 and line 314: Please cite Maffei et al (first systematic diverse phages isolated on a single host, BASEL collection and recent improvement) and recent use of barcode based tnsseq (for id of host factors and receptors) papers from Berkeley team PMID: 33048924 and 38048319 to provide a holistic view of recent publications in the field of phage characterization.
- Please cite prior work introduction, PMID: 16345976 which showed the importance of mixed indicator strain in isolating phages.

We are grateful for the reviewer's thoughtful comments which have allowed us to improve several aspects of the manuscript. We have responded to each reviewer comment below. Please note the line numbers in the following responses refer to the clean, non-marked up version of the manuscript file.

Reviewer 1:

Manuscript title: *Phage DisCo: targeted discovery of bacteriophages by co-culture*

Authors: Eleanor A. Rand, Natalia Quinones-Olvera, Kesther D. C. Jean, Carmen Hernandez Perez, Siân V. Owen, Michael Baym

Overall comment:

The authors build on their prior published (Phage Disco, PMID: 36993299) method to differentially identify phages based on their host-receptor utility for infection, and with host-defense sensitivity based on plaque assays involving co-cultures of bacterial strains with each strain carrying a fluorescent tag. This paper is well written and provides clear figures and data to present Phage Discovery by Co-culture (or Phage DisCo) method fitting the journal scope. We need improved methods for sampling phage diversity we observe by genomics. So this paper definitely provides a new handle towards this goal. However, claiming this method will solve all of our challenges in isolating novel phages and help in sampling phage diversity from an environment is far-fetched. Authors agree that there are obvious limitations to the method (target host needs to be engineerable etc), but claim that this method is suitable for surveying "ecological relationships across environments" Line 278. We recommend authors to either address this by detailing how they would go about building ecological relationships across environments and sample phage diversity or tone down their claims.

Here we provide the following comments and hope these comments will help authors to improve their report.

1. Based on the engineering needed for this method to work, we believe this method is best limited to model strains such as *E coli*/Salmonella. Extending this to diverse environmental bacterial isolates is unrealistic because of the unavailability of genetic tools (plasmids, promoter-RBS parts, reporters, selection markers, gene deletion methods). There is also a limit on the number of bacterial cultures that can be mixed in the co-cultures for the multiplex mode. Another point that needs consideration is potential co-culture interactions (e.g. growth promoting and/or inhibitory activity of one strain on another) and how they impact which phages can be isolated for the different strains. Anyways, Isolating phages on E coli is not going to provide us a catalog of phage diversity in the environment. For example, using a traditional phage isolation method using a single bacterial strain this report (PMID: 34784345) isolated highly diverse coliphages.
 - a. We agree that the phage disco approach is easiest to adopt with genetically tractable model organisms such as *E. coli* and *Salmonella*, or Gram positive models like *B. subtilis*, as discussed on lines 335-337. However, techniques for manipulating non model organisms exist and are constantly increasing, and therefore we do not consider this technique limited to model strains. We have added ideas and citations for how to manipulate diverse species to the text on lines 311-314.
 - b. We have addressed the number of strains that we believe could be combined when multiplexing Phage DisCo in the newly added paragraph beginning on 293. Although we have not tested greater than three strains, we believe with the right imaging set up, as many as 7 strains could be co-cultured together. As we are proposing that the cocultured strains in question would be isogenic, we don't anticipate antagonistic growth interactions between them.

- c. We agree that diverse phages can be isolated using a single bacterial strain. Rather than achieve general diversity, the goal of our approach is to isolate a specific phage or populations of phages of interest. We have toned down our claims accordingly throughout the manuscript.
2. We do not agree (for reasons stated in #1 above) that Phage Disco can “quickly survey an environment of both axes of diversity” line 287. The authors predominantly use relatively well characterized hosts and phages. However, for the vast majority of phages, not much is known about what host genes or genomic regions are important for phage infection (let alone, rare phages which the paper's method claims to be useful in isolating and characterizing). Furthermore, given the vast number of genes or genomic regions involved in phage infection, co-interactions, polarity that may be involved with the genes, etc. a significantly larger number of plaque assays/plates would need to be utilized to understand phage infection mechanisms using this method, potentially reducing throughput.
 - a. We see how the sentence previous on line 287 was misleading and have reworded it. Our intention is to convey that the same sample can be screened for both receptor specific phages as well as defense system specific phages by using different sets of indicator strains, to overcome the screening problem where the majority of phages in any sample are abundant and common. This has been clarified at lines 277-282.
 - b. Although we use characterized phages as controls to show that the method works as described, we hope that Phage DisCo will help discover new and specialized phages from the environment.
 - c. We indicate in the limitations paragraph at lines 320-323 that Phage DisCo is not intended to provide a broad picture of host genes required for phage infection. For this goal, an unbiased screening approach such as INSeq would be needed.
3. Even if the *Phage Disco* method makes it easy to identify phages that use a defined receptor, it is still necessary to isolate and sequence the genome of those phages to truly know the phage diversity. In that case, we don't think traditional culture-based method for phage discovery will be longer than designing an engineered host (by deleting a gene and developing fluorescent tag plasmid that works in that target host) compatible with the Phage Disco method. By using a 96 well plate Viral gDNA extraction kits and based on the average numbers mentioned line 184, at least one of each plaques identified as TolC-, Tsx- or PqqU-dependent in the environmental sample assay, could be identified.
 - a. The goal of the rarefaction analysis was to show that as phages of interest get rarer the relative benefits of methods like Phage DisCo increase. Depending on a lab's relative comfort with engineering bacterial strains and high throughput phage characterization, the cut off for when Phage DisCo becomes most useful may be different.

Sequencing phages does not generally allow identification of their receptors, particularly if they are uncharacterized phages. The goal of Phage DisCo is to rapidly identify phages with phenotypes of interest before further characterization and sequencing. This is especially helpful if the phage of interest is rare as has been clarified on lines 177-183.
 - b. We recommend that phages isolated using Phage DisCo are purified, replicated, and sequenced for full characterization. A sentence has been added on lines 264-266 to clarify this.
4. Given the fact that fluorescent markers need to be genetically expressed in the (model) strains, what is the maximum limit on the number of strains that can be co-cultured with

marker, such that fluorescence detection can be used to unambiguously identify plaques and what host strains they are lysing based on different colors/mixed colors?

We have added a paragraph beginning at 293 discussing the limitations of combining many strains in the co-culture lawn. The largest limitation is the availability of spectrally distinct fluorescent proteins, which is currently around seven.

5. The 46 Tsx-dependent plaques out of 746 representing 6% of all the screened plaques doesn't seem rare, especially because the entire diversity of the 746 plaques is not known. Moreover, are the plaques with a red halo shown in Figure 2A PqqU-dependent plaques? In that case the number of these plaques is bigger than 11 making those plaques less rare. The authors should provide information concerning those plaques to make a claim of diversity.
 - a. We have added clarity to lines 177-183 to explain that the efficiency gained by using Phage DisCo increases as phages become rarer. We believe this is an important point regardless of the rarity of the phages that we looked for in this example experiment.
 - b. We do not expect the plaques with red halos to be pqqU dependent and have added a sentence at lines 167-169 as well as Supp. Figure 4 to clarify.
6. We are intrigued by the false negative data on CBASS expt (line 241). We wonder if that may be because the CBASS system is expressed from a plasmid, and the overexpression of CBASS is causing this false negative effect. Authors should add clarification to this point in the last section of discussion (line 300-315).

Thank you for this suggestion. We tested dilutions of the inducer to see if lower levels of CBASS expression would bring out the red halo expected for a CBASS sensitive strain but there was no observable effect. These results are summarized on lines 237-238 and in Supp. Figure 6.

Minor comments

- Figure 1A: The petri dish illustration should be explained in the legend. I believe authors want to show that phage plaque morphology can be an inconsistent metric for differentiating phages, but it is not clear.

Thank you for drawing our attention to this. A description of the cartoon screening plate has been added to the figure legend for Figure 1A – lines 650-652.
- Line 64 and line 314: Please cite Maffei et al (first systematic diverse phages isolated on a single host, BASEL collection and recent improvement) and recent use of barcode based tseq (for id of host factors and receptors) papers from Berkeley team PMID: 33048924 and 38048319 to provide a holistic view of recent publications in the field of phage characterization.

We have cited the Berkeley team papers at lines 64 and 326-327.
- Please cite prior work introduction, PMID: 16345976 which showed the importance of mixed indicator strain in isolating phages.

Thank you for this suggestion, we have added a citation to this article at line 81.

Reviewer 2:

Peer review for “Phage DisCo: targeted discovery of bacteriophages by co-culture”

Overview: Rand and colleagues describe a new method and proof-of-concept study for the rapid and high-throughput identification of environmental phages. In this manuscript, they innovatively devise a technique combining bacterial molecular-biology, fluorescent-based imaging, and traditional aspects of bacteriophage biology to screen for specific phage phenotypes. Screenable phages include those that: a) putatively use specific receptors, or b)

are targeted by specific cytoplasmic resistance mechanisms. The authors validate the method with several phages, present cohesive data, and discuss exciting potential applications as well as some limitations of the method. The colorful composite fluorescent images evoke 20th-century-era discotheques. Overall, the manuscript is well-written, the rationale for the work is sound, and the science advances the fields of phage discovery and environmental microbiology. My major comments have to do with some aspects of the writing, clarity for some of the ideas presented, incomplete methodology, and structure of the manuscript. I think most of the comments can likely be addressed without additional data collection. Specific comments in order of appearance. The more major comments are highlighted by **bold** type.

1. **The Introduction lacks details on background of the study systems. (In some instances, this information is mis-placed in the Results, Methods, or Figure Legends.) Specifically, it would help to provide background on the phages used, their receptors, their lifecycles (lytic or temperate), and molecular mechanisms of the phage defense systems used.**

We have added a summary of the characterized phages, receptors, and defense systems that will be used later in the paper at Lines 84-90 of the introduction.

2. Results section: This commonly includes material that is not results:
 - a. Paragraph 1 contains a mix of Introductory material and Discussion material
 - b. Paragraph 2 is a mix of Methods and Discussion.

We have modified the section titles to better represent their content. We have changed “Results” and “Conclusion and Discussion” to “Results and Discussion” and “Conclusion”. While we agree that the boundaries of these sections are not perfectly discrete, we believe that some additional background in the results section is necessary for the narrative of the manuscript.

3. **Results for the Rank Abundance Curve (Fig. 1): No methods are provided for the simulated data in Fig. 1A. (Modeling technique, parameterization, software, etc.)**

Thank you for pointing out the lack of clarity for Figure 1A. We have added text to the figure legend at lines 649-650 clarifying that this graph is a cartoon to illustrate what the shape of the curve is meant to be rather than a true simulation.

4. Throughout the text, “clear plaque” should technically be “colorless plaque image” and “colored plaque” should be “fluorescent plaque” or “colored plaque image.” (In many areas of phage biology, the term “clear” plaque is used instead to describe very low plaque turbidity, an important component of plaque morphology.)

We agree that “clear” plaque might be confusing in the context of lysogenic phages. We have changed the term to “dark” plaque which is a good contrast to the “fluorescent” plaques caused by the differential plaquing of phages of interest.

5. Figure 1 provides a useful and easy-to-follow overview of the method.
6. Figure 2 and S2: beautiful plaques and overall figures.

Thank you for these kind comments

7. Throughout the text: It would be useful to, in the main text, name the phages used during validation.

The phage names have been added on lines 84-88 and Line 142.

8. **One limitation to the work is that the newly-discovered phages were not brought through a standard double-isolation/plaque purification. Were genomics or other methods used to test for plaque purity?**

- a. This is an important point, and we recommend that phages be purified, replicated, and fully characterized before further use. We have added a sentence to clarify this on lines 264-266.

- b. Additionally, our second step of plating the phages on each monoculture lawn gives an indication of purity, as plaquing is completely eliminated on the knockout strain of the corresponding receptor for each phage (Supp. Figure 3) though we note that contamination with phages dependent on the same target receptor would not be detectable.
9. **For the GmrSD experiment, one phage isolate had an unexpected genotype. Might this have actually been caused by a mixed phage sample? (See comment above about plaque purification.)**
 - a. We have added a sentence to clarify that phages must be further purified on lines 264-266.
 - b. We do not think that the CBASS false negative could be explained by contamination: if that stock was contaminated with a phage that CBASS could defend against, we would still see full plaquing ability on the CBASS⁺ strain. We have added a sentence to clarify this on lines 235-237.
10. Line 271-273: It would be more accurate to clarify that the method worked in only 2/3 cases, for example by adding something like the text shown in underline: “We also validated that the method can distinguish, in some cases, between control and environmental phages that interact with a given defense system...”

For all three cases Phage DisCo was able to distinguish between control phages that are sensitive/insensitive to the defense systems. Given that the controls worked and that we did not find phages in the environment that interact with the BstA system, this probably reflects the Salmonella phage population in this environment, rather than a failure of the method. We have clarified the sentence on lines 260-262.
11. **Line 274-275: “Phage DisCo can substantially reduce time and labor needed for targeted discovery of lytic or lysogenic phages with DNA or RNA genomes.” The manuscript presents no data on the lifestyle (lytic or lysogenic) or genomic structure (DNA or RNA) of any of the newly-isolated phages. This line should either be omitted, or the relevant phenotypic/genotypic data added.**

This sentence was in reference to the set of control phages used in this study as well as our previous study using phage disco. To avoid any confusion, we have removed this claim.
12. Lines 286 – 289: “Phage DisCo can quickly survey an environment for both axes of diversity and could even be nested to find phages with very specific properties...” No data in the manuscript support that this has been done, so I suggest revising the wording as a suggested application, rather than claiming this method *can* do this.

This sentence has been reworded to explain that the same environment can be screened for receptor specific or defense system specific phages using different sets of indicator strains. New wording on lines 277-282 hopefully makes this clear.
13. **Lines 291-295 - Writing in this passage is unclear, so it cannot be reliably peer-reviewed. Please revise the grammar, run-on sentence structure, and scientific term usage.**
 - a. “Here we have shown receptor knockouts and defense system knock-ins in lab strains of *E. coli*, but modifying environmental strains may help discover more environmentally relevant phages. For example, instead of heterologously expressing a defense system in *E. coli* wildtype bacterial isolates containing defense systems together with isogenic knockouts could allow more physiologically-relevant screens, potentially increasing positive hit rate.”

Text at lines 284-288 have been reworded for clarity.

14. **Lines 296-298. This section lacks clarity, and it is unclear how the proposed idea would work.**
 - a. “Furthermore, the use of Phage DisCo to isolate defense system-sensitive phages has the potential to provide insights into the molecular mechanisms of these systems themselves, as the evolution of resistant “escape” phages can pin-point phages proteins that trigger their activity [32].”
We have added clarification to this sentence on line 290, and more detail can be found in the accompanying reference.
15. Paragraph at Line 300 (“Although we anticipate...”). This is a nice summary of limitations.
Thank you!
16. Lines 309-314. No references are provided for several statements about phage receptors, LPS, etc.
References have been added on line 323.
17. Lines 314-315. “In this case, follow up, for example using the INSeq method [8], would needed to verify the full dependencies of any given phage.” Is INSeq necessary or even better than traditional methods? For example, could this instead be accomplished with generation of knockouts, complementation tests for plaque infectivity, host lysis, and/or phage adsorption?
The wording of this sentence on lines 323-327 has been updated to include more techniques and citations for characterizing phages.
18. **Line 346: Wastewater collection and processing: Please provide sample site data (environmental type, latitude/longitude coordinates, etc.).**
More detail on the sample type has been added on lines 361-362. We are prohibited from giving exact coordinates for the wastewater treatment plant but have included in the manuscript the maximum details allowed by the terms under which the samples were provided.
19. Line 408 - “Complement” assays should be “complementation assays.” (Complement assays are a method of immunology)
Thank you for pointing this out! All instances of complement assay have been changed to complementation assay.
20. Line 413: Monarch kit product # and supplier needed.
The NEB kit product number was added on line 426.
21. Line 422 in Methods section: “The dilutions in the raw data shown in Supp. Figure 3 represent the -3 through -10 dilutions.” This information should be in the figure labels or legend instead.
The sentence previously on line 422 was removed and we have confirmed that this information is available in the figure legends.
22. Line 472: For patchy/grainy lawns, this is a neat tip. Do you have data, a reference, or hypothesis for why this works?
We have added a citation for log phase growth on line 487 as well as data for more concentrated cells in Supp. Figure 8. We believe graininess is caused by areas of clonal cell expansion, and these areas can be reduced in size by seeding a greater number of cells into the lawn.
23. Line 611, Supp Table 2, Parts list. Rather than hyperlinking, add supplier and product # here (e.g., Luxeon star LEDs, SP-08-L1). This will help ensure long-term availability to the specific parts, which is especially important for this custom set-up.
All suppliers and product numbers were added to Supp. Table 2 to help with longer-term availability.
24. Line 616 and elsewhere: Italics for bacterial species names.
We have made sure italics are used for all bacterial species names

25. Line 626 Fig S3 labels: Instead of (or in addition to) “green, blue, and red”, it would be useful to label how the colors correspond to the gene knockouts.
Thank you for the suggestion, these labels have been changed.
26. Fig S5 legend. Much of this content would be more appropriately placed into the Background or Discussion sections:
- a. “BstA, which is encoded by the temperate phage BTP1, acts as a defense system against phages invading the bacterial host in which it has integrated. However, when BTP1 excises from the chromosome to lyse its host cell, it has an anti-defense region that allows for its life cycle to continue unimpeded. For this reason, BTP1 is expected to be able to lyse a cell with BstA on the chromosome while BTP1 Δ bstA is not [29]. The blue strain contains a BstA sequence with an early stop codon while the red contains a functioning BstA defense system. As is anticipated, BTP1 makes a clear plaque while BTP1 Δ bstA has red fluorescent signal within the plaque.”
We agree that this information is not required in the figure legend and have removed a lot of this information from the figure legend of Supp. Figure 6. More details on the defense system can be found in the corresponding citation.
27. You might consider naming or systematically numbering your newly-discovered phages. That would help future work on these isolates reference back to this manuscript.
As this manuscript focuses on the method rather than the biology of these phages, we have elected not to name or fully characterize the phages in this paper.
28. References: Typos and formatting throughout.
We have fixed all typos we could find in the references and updated them to ASM style format.
29. To the students: It is great to read a grad student’s first, first-author manuscript and to see that undergrads are co-authors as well. We reviewers know you put in many long days, generated a whole lot of data that are not in this manuscript, and conducted countless protocol development assays that aren’t even in the supplement. Those are the days and the data where a lot of science happens. You are doing excellent work. I look forward to future characterization of these new phages and seeing where Phage DisCo goes next.
Thank you, it’s been a long journey to get here.

Re: mSystems01644-24R1 (Phage DisCo: targeted discovery of bacteriophages by co-culture)

Dear Dr. Michael Baym:

Revision Guidelines

- Upload point-by-point responses to the issues raised by the reviewers in a file named "Response to Reviewers," NOT in your cover letter.
- Upload a compare copy of the manuscript (without figures) as a "Marked-Up Manuscript" file.
- Upload a clean .DOC/.DOCX version of the revised manuscript and remove the previous version.
- Each figure must be uploaded as a separate, editable, high-resolution file (TIFF or EPS preferred), and any multipanel figures must be assembled into one file.

Minireviews are not subject to publication charges.

Author Bios: We encourage you to submit a biographical sketch of each author (limit of 150 words) along with a photo to be published at the end of your article. You can submit these with your modified manuscript.

Figures Enhancement: ASM has engaged a professional science illustrator, Patrick Lane of ScEYence Studios, to work with minireview authors at the modification stage to generate improved figures that are uniform throughout the journal. This art enhancement service is free of charge to authors of minireviews and full-length reviews, and turnaround time is fast. I think you will be pleased with the results. Please contact Patrick on receiving this letter. Complete contact information for Patrick and further instructions are posted at <https://journals.asm.org/pb-assets/pdf-text-excel-files/graphical-enhancement-support.pdf>.

Sincerely,
Alejandro Reyes Munoz
Editor
mSystems

Reviewer #1 (Comments for the Author):

Authors have addressed all of our concerns and we are satisfied with all other editorial edits in this revised manuscript.

Reviewer #2 (Comments for the Author):

Most, but not all, of the comments have been addressed. See attached responses for comments 13, 14, and 22 that need further work, plus additional comments (8, 18, and 24) for your consideration.

We are grateful for the reviewer's thoughtful comments which have allowed us to improve several aspects of the manuscript. We have responded to each reviewer comment below. Please note the line numbers in the following responses refer to the clean, non-marked up version of the manuscript file.

Reviewer 1:

Manuscript title: *Phage DisCo: targeted discovery of bacteriophages by co-culture*

Authors: Eleanor A. Rand, Natalia Quinones-Olvera, Kesther D. C. Jean, Carmen Hernandez Perez, Siân V. Owen, Michael Baym

Overall comment:

The authors build on their prior published (Phage Disco, PMID: 36993299) method to differentially identify phages based on their host-receptor utility for infection, and with host-defense sensitivity based on plaque assays involving co-cultures of bacterial strains with each strain carrying a fluorescent tag. This paper is well written and provides clear figures and data to present Phage Discovery by Co-culture (or Phage DisCo) method fitting the journal scope. We need improved methods for sampling phage diversity we observe by genomics. So this paper definitely provides a new handle towards this goal. However, claiming this method will solve all of our challenges in isolating novel phages and help in sampling phage diversity from an environment is far-fetched. Authors agree that there are obvious limitations to the method (target host needs to be engineerable etc), but claim that this method is suitable for surveying "ecological relationships across environments " Line 278. We recommend authors to either address this by detailing how they would go about building ecological relationships across environments and sample phage diversity or tone down their claims.

Here we provide the following comments and hope these comments will help authors to improve their report.

1. Based on the engineering needed for this method to work, we believe this method is best limited to model strains such as *E. coli*/Salmonella. Extending this to diverse environmental bacterial isolates is unrealistic because of the unavailability of genetic tools (plasmids, promoter-RBS parts, reporters, selection markers, gene deletion methods). There is also a limit on the number of bacterial cultures that can be mixed in the co-cultures for the multiplex mode. Another point that needs consideration is potential co-culture interactions (e.g. growth promoting and/or inhibitory activity of one strain on another) and how they impact which phages can be isolated for the different strains. Anyways, Isolating phages on *E. coli* is not going to provide us a catalog of phage diversity in the environment. For example, using a traditional phage isolation method using a single bacterial strain this report (PMID: 34784345) isolated highly diverse coliphages.
 - a. We agree that the phage disco approach is easiest to adopt with genetically tractable model organisms such as *E. coli* and *Salmonella*, or Gram positive models like *B. subtilis*, as discussed on lines 335-337. However, techniques for manipulating non model organisms exist and are constantly increasing, and therefore we do not consider this technique limited to model strains. We have added ideas and citations for how to manipulate diverse species to the text on lines 311-314.
 - b. We have addressed the number of strains that we believe could be combined when multiplexing Phage DisCo in the newly added paragraph beginning on 293. Although we have not tested greater than three strains, we believe with the right imaging set up, as many as 7 strains could be co-cultured together. As we are proposing that the cocultured strains in question would be isogenic, we don't anticipate antagonistic growth interactions between them.

- c. We agree that diverse phages can be isolated using a single bacterial strain. Rather than achieve general diversity, the goal of our approach is to isolate a specific phage or populations of phages of interest. We have toned down our claims accordingly throughout the manuscript.
- 2. We do not agree (for reasons stated in #1 above) that Phage Disco can “quickly survey an environment of both axes of diversity” line 287. The authors predominantly use relatively well characterized hosts and phages. However, for the vast majority of phages, not much is known about what host genes or genomic regions are important for phage infection (let alone, rare phages which the paper’s method claims to be useful in isolating and characterizing). Furthermore, given the vast number of genes or genomic regions involved in phage infection, co-interactions, polarity that may be involved with the genes, etc. a significantly larger number of plaque assays/plates would need to be utilized to understand phage infection mechanisms using this method, potentially reducing throughput.
 - a. We see how the sentence previous on line 287 was misleading and have reworded it. Our intention is to convey that the same sample can be screened for both receptor specific phages as well as defense system specific phages by using different sets of indicator strains, to overcome the screening problem where the majority of phages in any sample are abundant and common. This has been clarified at lines 277-282.
 - b. Although we use characterized phages as controls to show that the method works as described, we hope that Phage DisCo will help discover new and specialized phages from the environment.
 - c. We indicate in the limitations paragraph at lines 320-323 that Phage DisCo is not intended to provide a broad picture of host genes required for phage infection. For this goal, an unbiased screening approach such as INSeq would be needed.
- 3. Even if the *Phage Disco* method makes it easy to identify phages that use a defined receptor, it is still necessary to isolate and sequence the genome of those phages to truly know the phage diversity. In that case, we don’t think traditional culture-based method for phage discovery will be longer than designing an engineered host (by deleting a gene and developing fluorescent tag plasmid that works in that target host) compatible with the Phage Disco method. By using a 96 well plate Viral gDNA extraction kits and based on the average numbers mentioned line 184, at least one of each plaques identified as TolC-, Tsx- or PqqU-dependent in the environmental sample assay, could be identified.
 - a. The goal of the rarefaction analysis was to show that as phages of interest get rarer the relative benefits of methods like Phage DisCo increase. Depending on a lab’s relative comfort with engineering bacterial strains and high throughput phage characterization, the cut off for when Phage DisCo becomes most useful may be different.

Sequencing phages does not generally allow identification of their receptors, particularly if they are uncharacterized phages. The goal of Phage DisCo is to rapidly identify phages with phenotypes of interest before further characterization and sequencing. This is especially helpful if the phage of interest is rare as has been clarified on lines 177-183.
 - b. We recommend that phages isolated using Phage DisCo are purified, replicated, and sequenced for full characterization. A sentence has been added on lines 264-266 to clarify this.
- 4. Given the fact that fluorescent markers need to be genetically expressed in the (model) strains, what is the maximum limit on the number of strains that can be co-cultured with

marker, such that fluorescence detection can be used to unambiguously identify plaques and what host strains they are lysing based on different colors/mixed colors?

We have added a paragraph beginning at 293 discussing the limitations of combining many strains in the co-culture lawn. The largest limitation is the availability of spectrally distinct fluorescent proteins, which is currently around seven.

5. The 46 Tsx-dependent plaques out of 746 representing 6% of all the screened plaques doesn't seem rare, especially because the entire diversity of the 746 plaques is not known. Moreover, are the plaques with a red halo shown in Figure 2A PqqU-dependent plaques? In that case the number of these plaques is bigger than 11 making those plaques less rare. The authors should provide information concerning those plaques to make a claim of diversity.
 - a. We have added clarity to lines 177-183 to explain that the efficiency gained by using Phage DisCo increases as phages become rarer. We believe this is an important point regardless of the rarity of the phages that we looked for in this example experiment.
 - b. We do not expect the plaques with red halos to be pqqU dependent and have added a sentence at lines 167-169 as well as Supp. Figure 4 to clarify.
6. We are intrigued by the false negative data on CBASS expt (line 241). We wonder if that may be because the CBASS system is expressed from a plasmid, and the overexpression of CBASS is causing this false negative effect. Authors should add clarification to this point in the last section of discussion (line 300-315).

Thank you for this suggestion. We tested dilutions of the inducer to see if lower levels of CBASS expression would bring out the red halo expected for a CBASS sensitive strain but there was no observable effect. These results are summarized on lines 237-238 and in Supp. Figure 6.

Minor comments

- Figure 1A: The petri dish illustration should be explained in the legend. I believe authors want to show that phage plaque morphology can be an inconsistent metric for differentiating phages, but it is not clear.

Thank you for drawing our attention to this. A description of the cartoon screening plate has been added to the figure legend for Figure 1A – lines 650-652.
- Line 64 and line 314: Please cite Maffei et al (first systematic diverse phages isolated on a single host, BASEL collection and recent improvement) and recent use of barcode based tseq (for id of host factors and receptors) papers from Berkeley team PMID: 33048924 and 38048319 to provide a holistic view of recent publications in the field of phage characterization.

We have cited the Berkeley team papers at lines 64 and 326-327.
- Please cite prior work introduction, PMID: 16345976 which showed the importance of mixed indicator strain in isolating phages.

Thank you for this suggestion, we have added a citation to this article at line 81.

Reviewer 2:

Peer review for “Phage DisCo: targeted discovery of bacteriophages by co-culture”

Overview: Rand and colleagues describe a new method and proof-of-concept study for the rapid and high-throughput identification of environmental phages. In this manuscript, they innovatively devise a technique combining bacterial molecular-biology, fluorescent-based imaging, and traditional aspects of bacteriophage biology to screen for specific phage phenotypes. Screenable phages include those that: a) putatively use specific receptors, or b)

are targeted by specific cytoplasmic resistance mechanisms. The authors validate the method with several phages, present cohesive data, and discuss exciting potential applications as well as some limitations of the method. The colorful composite fluorescent images evoke 20th-century-era discotheques. Overall, the manuscript is well-written, the rationale for the work is sound, and the science advances the fields of phage discovery and environmental microbiology. My major comments have to do with some aspects of the writing, clarity for some of the ideas presented, incomplete methodology, and structure of the manuscript. I think most of the comments can likely be addressed without additional data collection. Specific comments in order of appearance. The more major comments are highlighted by **bold** type.

1. **The Introduction lacks details on background of the study systems. (In some instances, this information is mis-placed in the Results, Methods, or Figure Legends.) Specifically, it would help to provide background on the phages used, their receptors, their lifecycles (lytic or temperate), and molecular mechanisms of the phage defense systems used.**

We have added a summary of the characterized phages, receptors, and defense systems that will be used later in the paper at Lines 84-90 of the introduction.
Addressed.

2. Results section: This commonly includes material that is not results:
 - a. Paragraph 1 contains a mix of Introductory material and Discussion material
 - b. Paragraph 2 is a mix of Methods and Discussion.

We have modified the section titles to better represent their content. We have changed “Results” and “Conclusion and Discussion” to “Results and Discussion” and “Conclusion”. While we agree that the boundaries of these sections are not perfectly discrete, we believe that some additional background in the results section is necessary for the narrative of the manuscript.
Addressed.

3. **Results for the Rank Abundance Curve (Fig. 1): No methods are provided for the simulated data in Fig. 1A. (Modeling technique, parameterization, software, etc.)**

Thank you for pointing out the lack of clarity for Figure 1A. We have added text to the figure legend at lines 649-650 clarifying that this graph is a cartoon to illustrate what the shape of the curve is meant to be rather than a true simulation.
Addressed.

4. Throughout the text, “clear plaque” should technically be “colorless plaque image” and “colored plaque” should be “fluorescent plaque” or “colored plaque image.” (In many areas of phage biology, the term “clear” plaque is used instead to describe very low plaque turbidity, an important component of plaque morphology.)

We agree that “clear” plaque might be confusing in the context of lysogenic phages. We have changed the term to “dark” plaque which is a good contrast to the “fluorescent” plaques caused by the differential plaquing of phages of interest.
Addressed.

5. Figure 1 provides a useful and easy-to-follow overview of the method.
6. Figure 2 and S2: beautiful plaques and overall figures.

Thank you for these kind comments
Addressed.

7. Throughout the text: It would be useful to, in the main text, name the phages used during validation.

The phage names have been added on lines 84-88 and Line 142.
Addressed.

8. **One limitation to the work is that the newly-discovered phages were not brought through a standard double-isolation/plaque purification. Were genomics or other methods used to test for plaque purity?**
 - a. This is an important point, and we recommend that phages be purified, replicated, and fully characterized before further use. We have added a sentence to clarify this on lines 264-266.
 - b. Addressed.
 - c.
 - d. Additionally, our second step of plating the phages on each monoculture lawn gives an indication of purity, as plaquing is completely eliminated on the knockout strain of the corresponding receptor for each phage (Supp. Figure 3) though we note that contamination with phages dependent on the same target receptor would not be detectable.
 - e. It's fine for this manuscript, but only because your goal was not to isolate and characterize phages. However, this is not solid reasoning in general. When plating, we typically only get on the order of 10^2 plaques. Contamination below 1% could well be common for phages (where typical stocks reach 10^7 to 10^{10} PFU/ml). Low-level contamination would not be detectable with phenotypic measures such as plaquing. Therefore, a minimum of double-isolation is still required when isolating and characterizing new phages, as seems to be a future direction of your work.
9. **For the GmrSD experiment, one phage isolate had an unexpected genotype. Might this have actually been caused by a mixed phage sample? (See comment above about plaque purification.)**
 - a. We have added a sentence to clarify that phages must be further purified on lines 264-266.
Addressed.
 - b. We do not think that the CBASS false negative could be explained by contamination: if that stock was contaminated with a phage that CBASS could defend against, we would still see full plaquing ability on the CBASS⁺ strain. We have added a sentence to clarify this on lines 235-237.
Addressed.
10. Line 271-273: It would be more accurate to clarify that the method worked in only 2/3 cases, for example by adding something like the text shown in underline: "We also validated that the method can distinguish, in some cases, between control and environmental phages that interact with a given defense system..."

For all three cases Phage DisCo was able to distinguish between control phages that are sensitive/insensitive to the defense systems. Given that the controls worked and that we did not find phages in the environment that interact with the BstA system, this probably reflects the Salmonella phage population in this environment, rather than a failure of the method. We have clarified the sentence on lines 260-262.
Addressed.
11. Line 274-275: "Phage DisCo can substantially reduce time and labor needed for targeted discovery of lytic or lysogenic phages with DNA or RNA genomes." The

manuscript presents no data on the lifestyle (lytic or lysogenic) or genomic structure (DNA or RNA) of any of the newly-isolated phages. This line should either be omitted, or the relevant phenotypic/genotypic data added.

This sentence was in reference to the set of control phages used in this study as well as our previous study using phage disco. To avoid any confusion, we have removed this claim.

Addressed.

12. Lines 286 – 289: “Phage DisCo can quickly survey an environment for both axes of diversity and could even be nested to find phages with very specific properties...” No data in the manuscript support that this has been done, so I suggest revising the wording as a suggested application, rather than claiming this method *can* do this.

This sentence has been reworded to explain that the same environment can be screened for receptor specific or defense system specific phages using different sets of indicator strains. New wording on lines 277-282 hopefully makes this clear.

Addressed.

13. **Lines 291-295 - Writing in this passage is unclear, so it cannot be reliably peer-reviewed. Please revise the grammar, run-on sentence structure, and scientific term usage.**

- a. “Here we have shown receptor knockouts and defense system knock-ins in lab strains of *E. coli*, but modifying environmental strains may help discover more environmentally relevant phages. For example, instead of heterologously expressing a defense system in *E. coli* wildtype bacterial isolates containing defense systems together with isogenic knockouts could allow more physiologically-relevant screens, potentially increasing positive hit rate.”

Text at lines 284-288 have been reworded for clarity.

Not addressed.

The text now has more clarity, but the proposed method is still not clear: “For example, instead of heterologously expressing a defense system in K-12 *E. coli*, pairing wildtype bacterial isolates containing defense systems together with isogenic knockouts could allow more physiologically-relevant screens, potentially increasing positive hit rate.”

Is the suggestion to engineer a single genome to have both a defense system and a knockout, or to have more than one bacterial genotype present in the screen? How would this change plaque morphology/color?

14. **Lines 296-298. This section lacks clarity, and it is unclear how the proposed idea would work.**

- a. “Furthermore, the use of Phage DisCo to isolate defense system-sensitive phages has the potential to provide insights into the molecular mechanisms of these systems themselves, as the evolution of resistant “escape” phages can pin-point phages proteins that trigger their activity [32].”

We have added clarification to this sentence on line 290, and more detail can be found in the accompanying reference.

Not addressed. The sentence remains the same, except for insertion of “and characterization,” which does not address the comment.

15. Paragraph at Line 300 (“Although we anticipate...”). This is a nice summary of limitations.
Thank you!
Addressed.
16. Lines 309-314. No references are provided for several statements about phage receptors, LPS, etc.
References have been added on line 323.
Addressed.
17. Lines 314-315. “In this case, follow up, for example using the INSeq method [8], would needed to verify the full dependencies of any given phage.” Is INSeq necessary or even better than traditional methods? For example, could this instead be accomplished with generation of knockouts, complementation tests for plaque infectivity, host lysis, and/or phage adsorption?
The wording of this sentence on lines 323-327 has been updated to include more techniques and citations for characterizing phages.
Addressed.
18. **Line 346: Wastewater collection and processing: Please provide sample site data (environmental type, latitude/longitude coordinates, etc.).**
More detail on the sample type has been added on lines 361-362. We are prohibited from giving exact coordinates for the wastewater treatment plant but have included in the manuscript the maximum details allowed by the terms under which the samples were provided.
That’s reasonable, and I think it would be great to include that reasoning (underlined text above) in the manuscript itself.
19. Line 408 - “Complement” assays should be “complementation assays.” (Complement assays are a method of immunology)
Thank you for pointing this out! All instances of complement assay have been changed to complementation assay.
Addressed
20. Line 413: Monarch kit product # and supplier needed.
The NEB kit product number was added on line 426.
Addressed
21. Line 422 in Methods section: “The dilutions in the raw data shown in Supp. Figure 3 represent the -3 through -10 dilutions.” This information should be in the figure labels or legend instead.
The sentence previously on line 422 was removed and we have confirmed that this information is available in the figure legends.
Addressed
22. Line 472: For patchy/grainy lawns, this is a neat tip. Do you have data, a reference, or hypothesis for why this works?
We have added a citation for log phase growth on line 487 as well as data for more concentrated cells in Supp. Figure 8. We believe graininess is caused by areas of clonal cell expansion, and these areas can be reduced in size by seeding a greater number of cells into the lawn.
Not addressed. The text has not been changed. (There is a tracked-change here, but the deleted text is the same as the “new” text.)

23. Line 611, Supp Table 2, Parts list. Rather than hyperlinking, add supplier and product # here (e.g., Luxeon star LEDs, SP-08-L1). This will help ensure long-term availability to the specific parts, which is especially important for this custom set-up.

All suppliers and product numbers were added to Supp. Table 2 to help with longer-term availability.

Addressed

24. Line 616 and elsewhere: Italics for bacterial species names.

We have made sure italics are used for all bacterial species names
Formatting issues remain in references section.

25. Line 626 Fig S3 labels: Instead of (or in addition to) “green, blue, and red”, it would be useful to label how the colors correspond to the gene knockouts.

Thank you for the suggestion, these labels have been changed.

Addressed

26. Fig S5 legend. Much of this content would be more appropriately placed into the Background or Discussion sections:

- a. “BstA, which is encoded by the temperate phage BTP1, acts as a defense system against phages invading the bacterial host in which it has integrated. However, when BTP1 excises from the chromosome to lyse its host cell, it has an anti-defense region that allows for its life cycle to continue unimpeded. For this reason, BTP1 is expected to be able to lyse a cell with BstA on the chromosome while BTP1 Δ bstA is not [29]. The blue strain contains a BstA sequence with an early stop codon while the red contains a functioning BstA defense system. As is anticipated, BTP1 makes a clear plaque while BTP1 Δ bstA has red fluorescent signal within the plaque.”

We agree that this information is not required in the figure legend and have removed a lot of this information from the figure legend of Supp. Figure 6. More details on the defense system can be found in the corresponding citation.

Addressed

27. You might consider naming or systematically numbering your newly-discovered phages. That would help future work on these isolates reference back to this manuscript.

As this manuscript focuses on the method rather than the biology of these phages, we have elected not to name or fully characterize the phages in this paper.

I suppose that's reasonable, although it might reduce how much your work is cited.

28. References: Typos and formatting throughout.

We have fixed all typos we could find in the references and updated them to ASM style format.

Mostly addressed, although see #24.

29. To the students: It is great to read a grad student's first, first-author manuscript and to see that undergrads are co-authors as well. We reviewers know you put in many long days, generated a whole lot of data that are not in this manuscript, and conducted countless protocol development assays that aren't even in the supplement. Those are the days and the data where a lot of science happens. You are doing excellent work. I look forward to future characterization of these new phages and seeing where Phage DisCo goes next.

Thank you, it's been a long journey to get here.

:)

We are grateful for the reviewer's thoughtful comments which have allowed us to improve several aspects of the manuscript. We have responded to each reviewer comment below. Please note the line numbers in the following responses refer to the clean, non-marked up version of the manuscript file.

Reviewer #1 (Comments for the Author):

Authors have addressed all of our concerns and we are satisfied with all other editorial edits in this revised manuscript.

Reviewer #2 (Comments for the Author):

Most, but not all, of the comments have been addressed. See attached responses for comments 13, 14, and 22 that need further work, plus additional comments (8, 18, and 24) for your consideration.

We have further addressed reviewer 2's remaining concerns. For completeness, we include the entire file of their comments the reviewer attached, however, all but comments 8, 13, 14, 18, 22, and 24 have been greyed out for ease of reading. We hope the updated manuscript now matches the reviewer's expectations.

Reviewer 2:

Peer review for "Phage DisCo: targeted discovery of bacteriophages by co-culture"

Overview: Rand and colleagues describe a new method and proof-of-concept study for the rapid and high-throughput identification of environmental phages. In this manuscript, they innovatively devise a technique combining bacterial molecular-biology, fluorescent-based imaging, and traditional aspects of bacteriophage biology to screen for specific phage phenotypes. Screenable phages include those that: a) putatively use specific receptors, or b) are targeted by specific cytoplasmic resistance mechanisms. The authors validate the method with several phages, present cohesive data, and discuss exciting potential applications as well as some limitations of the method. The colorful composite fluorescent images evoke 20th-century-era discotheques. Overall, the manuscript is well-written, the rationale for the work is sound, and the science advances the fields of phage discovery and environmental microbiology. My major comments have to do with some aspects of the writing, clarity for some of the ideas presented, incomplete methodology, and structure of the manuscript. I think most of the comments can likely be addressed without additional data collection. Specific comments in order of appearance. The more major comments are highlighted by **bold** type.

1. **The Introduction lacks details on background of the study systems. (In some instances, this information is mis-placed in the Results, Methods, or Figure Legends.) Specifically, it would help to provide background on the phages used, their receptors, their lifecycles (lytic or temperate), and molecular mechanisms of the phage defense systems used.**

We have added a summary of the characterized phages, receptors, and defense systems that will be used later in the paper at Lines 84-90 of the introduction. Addressed.

2. Results section: This commonly includes material that is not results:

- a. Paragraph 1 contains a mix of Introductory material and Discussion material
- b. Paragraph 2 is a mix of Methods and Discussion.

We have modified the section titles to better represent their content. We have changed “Results” and “Conclusion and Discussion” to “Results and Discussion” and “Conclusion”. While we agree that the boundaries of these sections are not perfectly discrete, we believe that some additional background in the results section is necessary for the narrative of the manuscript.
Addressed.

3. **Results for the Rank Abundance Curve (Fig. 1): No methods are provided for the simulated data in Fig. 1A. (Modeling technique, parameterization, software, etc.)**
Thank you for pointing out the lack of clarity for Figure 1A. We have added text to the figure legend at lines 649-650 clarifying that this graph is a cartoon to illustrate what the shape of the curve is meant to be rather than a true simulation.
Addressed.
4. Throughout the text, “clear plaque” should technically be “colorless plaque image” and “colored plaque” should be “fluorescent plaque” or “colored plaque image.” (In many areas of phage biology, the term “clear” plaque is used instead to describe very low plaque turbidity, an important component of plaque morphology.)
We agree that “clear” plaque might be confusing in the context of lysogenic phages. We have changed the term to “dark” plaque which is a good contrast to the “fluorescent” plaques caused by the differential plaquing of phages of interest.
Addressed.
5. Figure 1 provides a useful and easy-to-follow overview of the method.
6. Figure 2 and S2: beautiful plaques and overall figures.
Thank you for these kind comments
Addressed.
7. Throughout the text: It would be useful to, in the main text, name the phages used during validation.
The phage names have been added on lines 84-88 and Line 142.
Addressed.
8. **One limitation to the work is that the newly-discovered phages were not brought through a standard double-isolation/plaque purification. Were genomics or other methods used to test for plaque purity?**
 - a. This is an important point, and we recommend that phages be purified, replicated, and fully characterized before further use. We have added a sentence to clarify this on lines 264-266.
 - b. Addressed.
 - c. Additionally, our second step of plating the phages on each monoculture lawn gives in indication of purity, as plaquing is completely eliminated on the knockout strain of the corresponding receptor for each phage (Supp. Figure 3) though we note that contamination with phages dependent on the same target receptor would not be detectable.
 - d. It’s fine for this manuscript, but only because your goal was not to isolate and characterize phages. However, this is not solid reasoning in general. When plating, we typically only get on the order of 10^2 plaques. Contamination below 1% could well be common for phages (where typical stocks reach 10^7 to 10^{10} PFU/ml). Low-level contamination would not be detectable with phenotypic measures such as plaquing. Therefore, a minimum of double-isolation is still

required when isolating and characterizing new phages, as seems to be a future direction of your work.

- i. All phages were replicated before plating on monoculture lawns. As shown in the y-axis of Figure 2B and in the raw data in Supplemental Figure 3, every phage reached a titer of 10^8 or greater, and any contamination would have been replicated as well. Mathematically, this means that any contamination of a phage using a different receptor above $10^{-7}\%$ in the original plaque would be visible as plaques on the knockout strains shown in Supplemental Figure 3. As it is not, we can conclude that contamination is at highest ten million-fold lower than 1%.

9. **For the GmrSD experiment, one phage isolate had an unexpected genotype. Might this have actually been caused by a mixed phage sample? (See comment above about plaque purification.)**

- a. We have added a sentence to clarify that phages must be further purified on lines 264-266.

Addressed.

- b. We do not think that the CBASS false negative could be explained by contamination: if that stock was contaminated with a phage that CBASS could defend against, we would still see full plaquing ability on the CBASS⁺ strain We have added a sentence to clarify this on lines 235-237.

Addressed.

10. Line 271-273: It would be more accurate to clarify that the method worked in only 2/3 cases, for example by adding something like the text shown in underline: “We also validated that the method can distinguish, in some cases, between control and environmental phages that interact with a given defense system...”

For all three cases Phage DisCo was able to distinguish between control phages that are sensitive/insensitive to the defense systems. Given that the controls worked and that we did not find phages in the environment that interact with the BstA system, this probably reflects the Salmonella phage population in this environment, rather than a failure of the method. We have clarified the sentence on lines 260-262.

Addressed.

11. **Line 274-275: “Phage DisCo can substantially reduce time and labor needed for targeted discovery of lytic or lysogenic phages with DNA or RNA genomes.” The manuscript presents no data on the lifestyle (lytic or lysogenic) or genomic structure (DNA or RNA) of any of the newly-isolated phages. This line should either be omitted, or the relevant phenotypic/genotypic data added.**

This sentence was in reference to the set of control phages used in this study as well as our previous study using phage disco. To avoid any confusion, we have removed this claim.

Addressed.

12. Lines 286 – 289: “Phage DisCo can quickly survey an environment for both axes of diversity and could even be nested to find phages with very specific properties...” No data in the manuscript support that this has been done, so I suggest revising the wording as a suggested application, rather than claiming this method *can* do this.

This sentence has been reworded to explain that the same environment can be screened for receptor specific or defense system specific phages using different sets of indicator strains. New wording on lines 277-282 hopefully makes this clear.

Addressed.

13. **Lines 291-295 - Writing in this passage is unclear, so it cannot be reliably peer-reviewed. Please revise the grammar, run-on sentence structure, and scientific term usage.**

- a. “Here we have shown receptor knockouts and defense system knock-ins in lab strains of *E. coli*, but modifying environmental strains may help discover more environmentally relevant phages. For example, instead of heterologously expressing a defense system in *E. coli* wildtype bacterial isolates containing defense systems together with isogenic knockouts could allow more physiologically-relevant screens, potentially increasing positive hit rate.”

Text at lines 284-288 have been reworded for clarity.

Not addressed.

The text now has more clarity, but the proposed method is still not clear: “For example, instead of heterologously expressing a defense system in K-12 *E. coli*, pairing wildtype bacterial isolates containing defense systems together with isogenic knockouts could allow more physiologically-relevant screens, potentially increasing positive hit rate.”

Is the suggestion to engineer a single genome to have both a defense system and a knockout, or to have more than one bacterial genotype present in the screen? How would this change plaque morphology/color?

The suggestion here is to pair an environmental bacterial isolate with an isogenic partner in which one knockout has been introduced. This is the exact same set up that we used in the experiments here where the strains in co-culture are almost completely isogenic, but instead of introducing a defense system into a lab strain to create the pair, we propose knocking it out of a strain in which it is native. Using a more environmentally meaningful genetic background may help find more environmentally meaningful phages. For example, if you are interested in a medically relevant phage for treating human infections, using an infectious strain of *E. coli* is likely more useful than the benign K-12 strain. We have added this example for clarification on lines 288-291 and hope this now satisfies the reviewer.

14. **Lines 296-298. This section lacks clarity, and it is unclear how the proposed idea would work.**

- a. “Furthermore, the use of Phage DisCo to isolate defense system-sensitive phages has the potential to provide insights into the molecular mechanisms of these systems themselves, as the evolution of resistant “escape” phages can pin-point phages proteins that trigger their activity [32].”

We have added clarification to this sentence on line 290, and more detail can be found in the accompanying reference.

Not addressed. The sentence remains the same, except for insertion of “and characterization,” which does not address the comment.

We have reworded this section more thoroughly on lines 293-296.

15. Paragraph at Line 300 (“Although we anticipate...”). This is a nice summary of limitations.

Thank you!

Addressed.

16. Lines 309-314. No references are provided for several statements about phage receptors, LPS, etc.
References have been added on line 323.
Addressed.
17. Lines 314-315. “In this case, follow up, for example using the INSeq method [8], would needed to verify the full dependencies of any given phage.” Is INSeq necessary or even better than traditional methods? For example, could this instead be accomplished with generation of knockouts, complementation tests for plaque infectivity, host lysis, and/or phage adsorption?
The wording of this sentence on lines 323-327 has been updated to include more techniques and citations for characterizing phages.
Addressed.
18. **Line 346: Wastewater collection and processing: Please provide sample site data (environmental type, latitude/longitude coordinates, etc.).**
More detail on the sample type has been added on lines 361-362. We are prohibited from giving exact coordinates for the wastewater treatment plant but have included in the manuscript the maximum details allowed by the terms under which the samples were provided.
That’s reasonable, and I think it would be great to include that reasoning (underlined text above) in the manuscript itself.
This has been added on line 368.
19. Line 408 - “Complement” assays should be “complementation assays.” (Complement assays are a method of immunology)
Thank you for pointing this out! All instances of complement assay have been changed to complementation assay.
Addressed.
20. Line 413: Monarch kit product # and supplier needed.
The NEB kit product number was added on line 426.
Addressed.
21. Line 422 in Methods section: “The dilutions in the raw data shown in Supp. Figure 3 represent the -3 through -10 dilutions.” This information should be in the figure labels or legend instead.
The sentence previously on line 422 was removed and we have confirmed that this information is available in the figure legends.
Addressed.
22. Line 472: For patchy/grainy lawns, this is a neat tip. Do you have data, a reference, or hypothesis for why this works?
We have added a citation for log phase growth on line 487 as well as data for more concentrated cells in Supp. Figure 8. We believe graininess is caused by areas of clonal cell expansion, and these areas can be reduced in size by seeding a greater number of cells into the lawn.
Not addressed. The text has not been changed. (There is a tracked-change here, but the deleted text is the same as the “new” text.)
Apologies, the line number pointing to the reference was incorrect. The citation for log phages growth is on line 496, and the reference to Supplemental Figure 8 which has data on concentrating cells is on line 499.
23. Line 611, Supp Table 2, Parts list. Rather than hyperlinking, add supplier and product # here (e.g., Luxeon star LEDs, SP-08-L1). This will help ensure long-term availability to the specific parts, which is especially important for this custom set-up.

All suppliers and product numbers were added to Supp. Table 2 to help with longer-term availability.

Addressed.

24. Line 616 and elsewhere: Italics for bacterial species names.

We have made sure italics are used for all bacterial species names
Formatting issues remain in references section.

Italics has been added to reference 13, 22, 23, 28, 43, and 44.

25. Line 626 Fig S3 labels: Instead of (or in addition to) “green, blue, and red”, it would be useful to label how the colors correspond to the gene knockouts.

Thank you for the suggestion, these labels have been changed.

Addressed.

26. Fig S5 legend. Much of this content would be more appropriately placed into the Background or Discussion sections:

- a. “BstA, which is encoded by the temperate phage BTP1, acts as a defense system against phages invading the bacterial host in which it has integrated. However, when BTP1 excises from the chromosome to lyse its host cell, it has an anti-defense region that allows for its life cycle to continue unimpeded. For this reason, BTP1 is expected to be able to lyse a cell with BstA on the chromosome while BTP1 Δ bstA is not [29]. The blue strain contains a BstA sequence with an early stop codon while the red contains a functioning BstA defense system. As is anticipated, BTP1 makes a clear plaque while BTP1 Δ bstA has red fluorescent signal within the plaque.”

We agree that this information is not required in the figure legend and have removed a lot of this information from the figure legend of Supp. Figure 6. More details on the defense system can be found in the corresponding citation.

Addressed.

27. You might consider naming or systematically numbering your newly-discovered phages. That would help future work on these isolates reference back to this manuscript.

As this manuscript focuses on the method rather than the biology of these phages, we have elected not to name or fully characterize the phages in this paper.

I suppose that’s reasonable, although it might reduce how much your work is cited.

28. References: Typos and formatting throughout.

We have fixed all typos we could find in the references and updated them to ASM style format.

Mostly addressed, although see #24.

Italics has been added to reference 13, 22, 23, 28, 43, and 44.

29. To the students: It is great to read a grad student’s first, first-author manuscript and to see that undergrads are co-authors as well. We reviewers know you put in many long days, generated a whole lot of data that are not in this manuscript, and conducted countless protocol development assays that aren’t even in the supplement. Those are the days and the data where a lot of science happens. You are doing excellent work. I look forward to future characterization of these new phages and seeing where Phage DisCo goes next.

Thank you, it’s been a long journey to get here.

:)

Re: mSystems01644-24R2 (Phage DisCo: targeted discovery of bacteriophages by co-culture)

Dear Dr. Michael Baym:

Thanks for the responses to the reviewers comments. Although one of the reviewers still has some comments, I have gone through and consider they are not worth of another round of revisions. I am still attaching the reviewer response for your consideration. I have to say I agree with the reasoning of the reviewer in comment #8, although not relevant for the current manuscript. For comment #18, may I suggest replacing the parenthesis text for something along the lines of "Exact coordinates are not provided due to restrictions associated with the sample collection agreement".

Your manuscript has been accepted, and I am forwarding it to the ASM production staff for publication. Your paper will first be checked to make sure all elements meet the technical requirements. ASM staff will contact you if anything needs to be revised before copyediting and production can begin. Otherwise, you will be notified when your proofs are ready to be viewed.

Sincerely,
Alejandro Reyes Munoz
Editor
mSystems

Reviewer #1 (Comments for the Author):

Authors have addressed all of my comments

Reviewer #2 (Comments for the Author):

Please see attached.

We are grateful for the reviewer's thoughtful comments which have allowed us to improve several aspects of the manuscript. We have responded to each reviewer comment below. Please note the line numbers in the following responses refer to the clean, non-marked up version of the manuscript file.

Reviewer #1 (Comments for the Author):

Authors have addressed all of our concerns and we are satisfied with all other editorial edits in this revised manuscript.

Reviewer #2 (Comments for the Author):

Most, but not all, of the comments have been addressed. See attached responses for comments 13, 14, and 22 that need further work, plus additional comments (8, 18, and 24) for your consideration.

We have further addressed reviewer 2's remaining concerns. For completeness, we include the entire file of their comments the reviewer attached, however, all but comments 8, 13, 14, 18, 22, and 24 have been greyed out for ease of reading. We hope the updated manuscript now matches the reviewer's expectations.

Thank you for making this so organized and nice to read. Using the gray-out is a clever approach that nicely balances transparency and readability. My further comments are in green.

Reviewer 2:

Peer review for "Phage DisCo: targeted discovery of bacteriophages by co-culture"

Overview: Rand and colleagues describe a new method and proof-of-concept study for the rapid and high-throughput identification of environmental phages. In this manuscript, they innovatively devise a technique combining bacterial molecular-biology, fluorescent-based imaging, and traditional aspects of bacteriophage biology to screen for specific phage phenotypes. Screenable phages include those that: a) putatively use specific receptors, or b) are targeted by specific cytoplasmic resistance mechanisms. The authors validate the method with several phages, present cohesive data, and discuss exciting potential applications as well as some limitations of the method. The colorful composite fluorescent images evoke 20th-century-era discotheques. Overall, the manuscript is well-written, the rationale for the work is sound, and the science advances the fields of phage discovery and environmental microbiology. My major comments have to do with some aspects of the writing, clarity for some of the ideas presented, incomplete methodology, and structure of the manuscript. I think most of the comments can likely be addressed without additional data collection. Specific comments in order of appearance. The more major comments are highlighted by **bold** type.

1. The Introduction lacks details on background of the study systems. (In some instances, this information is mis-placed in the Results, Methods, or Figure Legends.) Specifically, it would help to provide background on the phages used, their receptors, their lifecycles (lytic or temperate), and molecular mechanisms of the phage defense systems used.

We have added a summary of the characterized phages, receptors, and defense systems that will be used later in the paper at Lines 84-90 of the introduction. Addressed.

2. Results section: This commonly includes material that is not results:**a.** Paragraph 1 contains a mix of Introductory material and Discussion material
b. Paragraph 2 is a mix of Methods and Discussion.

We have modified the section titles to better represent their content. We have changed "Results" and "Conclusion and Discussion" to "Results and Discussion" and "Conclusion". While we agree that the boundaries of these sections are not perfectly discrete, we believe that some additional background in the results section is necessary for the narrative of the manuscript.

Addressed.

3. Results for the Rank Abundance Curve (Fig. 1): No methods are provided for the simulated data in Fig. 1A. (Modeling technique, parameterization, software, etc.)

Thank you for pointing out the lack of clarity for Figure 1A. We have added text to

the figure legend at lines 649-650 clarifying that this graph is a cartoon to illustrate what the shape of the curve is meant to be rather than a true simulation. Addressed.

4. Throughout the text, “clear plaque” should technically be “colorless plaque image” and “colored plaque” should be “fluorescent plaque” or “colored plaque image.” (In many areas of phage biology, the term “clear” plaque is used instead to describe very low plaque turbidity, an important component of plaque morphology.)

We agree that “clear” plaque might be confusing in the context of lysogenic phages. We have changed the term to “dark” plaque which is a good contrast to the “fluorescent” plaques caused by the differential plaquing of phages of interest.

Addressed.

5. Figure 1 provides a useful and easy-to-follow overview of the method.

6. Figure 2 and S2: beautiful plaques and overall figures.

Thank you for these kind comments

Addressed.

7. Throughout the text: It would be useful to, in the main text, name the phages used during validation.

The phage names have been added on lines 84-88 and Line 142.

Addressed.

8. One limitation to the work is that the newly-discovered phages were not brought through a standard double-isolation/plaque purification. Were genomics or other methods used to test for plaque purity?

a. This is an important point, and we recommend that phages be purified, replicated, and fully characterized before further use. We have added a sentence to clarify this on lines 264-266.

b. Addressed.

c. Additionally, our second step of plating the phages on each monoculture lawn gives an indication of purity, as plaquing is completely eliminated on the knockout strain of the corresponding receptor for each phage (Supp. Figure 3) though we note that contamination with phages dependent on the same target receptor would not be detectable.

d. It's fine for this manuscript, but only because your goal was not to isolate and characterize phages. However, this is not solid reasoning in general. When plating, we typically only get on the order of 10^2 plaques. Contamination below 1% could well be common for phages (where typical stocks reach 10^7 to 10^{10} PFU/ml). Low-level contamination would not be detectable with phenotypic measures such as plaquing. Therefore, a minimum of double-isolation is still required when isolating and characterizing new phages, as seems to be a future direction of your work.

i. All phages were replicated before plating on monoculture lawns. As shown in the y-axis of Figure 2B and in the raw data in Supplemental Figure 3, every phage reached a titer of 10^8 or greater, and any contamination would have been replicated as well. Mathematically, this means that any contamination of a phage using a different receptor above 10⁻⁷% in the original plaque would be visible as plaques on the knockout strains shown in Supplemental Figure 3. As it is not, we can conclude that contamination is at highest ten million-fold lower than 1%.

That is not correct. Although a titer of a phage stock may be 10^8 PFU/ml, the samples titer on an agar plate is only about 100-200 plaques (10^2), representing just a tiny fraction of the phages in the stock. It is not possible to screen 10^8 plaques. That would take about a million agar plates. Again, it's okay for this manuscript, but this will continue to be an issue for any further work stemming from these non-plaque-purified phage stocks.

9. **For the GmrSD experiment, one phage isolate had an unexpected genotype. Might this have actually been caused by a mixed phage sample? (See comment above about plaque purification.)**

a. We have added a sentence to clarify that phages must be further purified on lines 264-266.

Addressed.

b. We do not think that the CBASS false negative could be explained by contamination: if that stock was contaminated with a phage that CBASS could defend against, we would still see full plaquing ability on the CBASS+ strain We have added a sentence to clarify this on lines 235-237.

Addressed.

10. Line 271-273: It would be more accurate to clarify that the method worked in only 2/3 cases, for example by adding something like the text shown in underline: “We also validated that the method can distinguish, in some cases, between control and environmental phages that interact with a given defense system...”

For all three cases Phage DisCo was able to distinguish between control phages that are sensitive/insensitive to the defense systems. Given that the controls worked and that we did not find phages in the environment that interact with the BstA system, this probably reflects the Salmonella phage population in this environment, rather than a failure of the method. We have clarified the sentence on lines 260-262.

Addressed.

11. **Line 274-275: “Phage DisCo can substantially reduce time and labor needed for targeted discovery of lytic or lysogenic phages with DNA or RNA genomes.” The manuscript presents no data on the lifestyle (lytic or lysogenic) or genomic structure (DNA or RNA) of any of the newly-isolated phages. This line should either be omitted, or the relevant phenotypic/genotypic data added.**

This sentence was in reference to the set of control phages used in this study as well as our previous study using phage disco. To avoid any confusion, we have removed this claim.

Addressed.

12. Lines 286 – 289: “Phage DisCo can quickly survey an environment for both axes of diversity and could even be nested to find phages with very specific properties...” No data in the manuscript support that this has been done, so I suggest revising the wording as a suggested application, rather than claiming this method *can* do this.

This sentence has been reworded to explain that the same environment can be screened for receptor specific or defense system specific phages using different sets of indicator strains. New wording on lines 277-282 hopefully makes this clear.

Addressed.13. **Lines 291-295 - Writing in this passage is unclear, so it cannot be reliably peer-reviewed. Please revise the grammar, run-on sentence structure, and scientific term usage.**

a. “Here we have shown receptor knockouts and defense system knock-ins in lab strains of *E. coli*, but modifying environmental strains may help discover more environmentally relevant phages. For example, instead of heterologously expressing a defense system in *E. coli* wildtype bacterial isolates containing defense systems together with isogenic knockouts could allow more physiologically-relevant screens, potentially increasing positive hit rate.”

Text at lines 284-288 have been reworded for clarity.

Not addressed.

The text now has more clarity, but the proposed method is still not clear: “For example, instead of heterologously expressing a defense system in K-12 *E. coli*, pairing wildtype bacterial isolates containing defense systems together with

isogenic knockouts could allow more physiologically-relevant screens, potentially increasing positive hit rate.”

Is the suggestion to engineer a single genome to have both a defense system and a knockout, or to have more than one bacterial genotype present in the screen? How would this change plaque morphology/color?

The suggestion here is to pair an environmental bacterial isolate with an isogenic partner in which one knockout has been introduced. This is the exact same set up that we used in the experiments here where the strains in co-culture are almost completely isogenic, but instead of introducing a defense system into a lab strain to create the pair, we propose knocking it out of a strain in which it is native. Using a more environmentally meaningful genetic background may help find more environmentally meaningful phages. For example, if you are interested in a medically relevant phage for treating human infections, using an infectious strain of *E. coli* is likely more useful than the benign K-12 strain. We have added this example for clarification on lines 288-291 and hope this now satisfies the reviewer.

Addressed.

14. Lines 296-298. This section lacks clarity, and it is unclear how the proposed idea would work.

a. “Furthermore, the use of Phage DisCo to isolate defense system-sensitive phages has the potential to provide insights into the molecular mechanisms of these systems themselves, as the evolution of resistant “escape” phages can pin-point phages proteins that trigger their activity [32].”

We have added clarification to this sentence on line 290, and more detail can be found in the accompanying reference.

Not addressed. The sentence remains the same, except for insertion of “and characterization,” which does not address the comment.

We have reworded this section more thoroughly on lines 293-296.

The new text reads: “For instance, once isolated using Phage DisCo, mutants of defense system-sensitive phages could be evolved by passaging through bacterial strains that contain the corresponding defense system and studied to pin-point how the phage and defense system interact (37).” Clarity is lacking and comprehension still difficult. Is the suggestion to evolve the *phages* or the *bacteria*? Is the idea to use phage mutation analysis to yield hypotheses about phage-host protein interactions?

15. Paragraph at Line 300 (“Although we anticipate...”). This is a nice summary of limitations.

Thank you!

Addressed. 16. Lines 309-314. No references are provided for several statements about phage receptors, LPS, etc.

References have been added on line 323.

Addressed.

17. Lines 314-315. “In this case, follow up, for example using the INSeq method [8], would be needed to verify the full dependencies of any given phage.” Is INSeq necessary or even better than traditional methods? For example, could this instead be accomplished with generation of knockouts, complementation tests for plaque infectivity, host lysis, and/or phage adsorption?

The wording of this sentence on lines 323-327 has been updated to include more techniques and citations for characterizing phages.

Addressed.

18. Line 346: Wastewater collection and processing: Please provide sample site data (environmental type, latitude/longitude coordinates, etc.).

More detail on the sample type has been added on lines 361-362. We are prohibited from giving exact coordinates for the wastewater treatment plant but have included in the manuscript the maximum details allowed by the terms under which the samples were provided.

That's reasonable, and I think it would be great to include that reasoning (underlined text above) in the manuscript itself.

This has been added on line 368.

I suggest using the language you used in the above response: "We are prohibited from giving exact coordinates for the wastewater treatment plant but have included in the manuscript the maximum details allowed by the terms under which the samples were provided." Otherwise, it is unclear who or what is prohibiting the detail (City ordinances? Your own lack of collecting these data? Terms of an agreement?)

19. Line 408 - "Complement" assays should be "complementation assays." (Complement assays are a method of immunology)

Thank you for pointing this out! All instances of complement assay have been changed to complementation assay.

Addressed.

20. Line 413: Monarch kit product # and supplier needed.

The NEB kit product number was added on line 426.

Addressed.

21. Line 422 in Methods section: "The dilutions in the raw data shown in Supp. Figure 3 represent the -3 through -10 dilutions." This information should be in the figure labels or legend instead.

The sentence previously on line 422 was removed and we have confirmed that this information is available in the figure legends.

Addressed.

22. Line 472: For patchy/grainy lawns, this is a neat tip. Do you have data, a reference, or hypothesis for why this works?

We have added a citation for log phase growth on line 487 as well as data for more concentrated cells in Supp. Figure 8. We believe graininess is caused by areas of clonal cell expansion, and these areas can be reduced in size by seeding a greater number of cells into the lawn.

Not addressed. The text has not been changed. (There is a tracked-change here, but the deleted text is the same as the "new" text.)

Apologies, the line number pointing to the reference was incorrect. The citation for log phages growth is on line 496, and the reference to Supplemental Figure 8 which has data on concentrating cells is on line 499.

Addressed.

23. Line 611, Supp Table 2, Parts list. Rather than hyperlinking, add supplier and product # here (e.g., Luxeon star LEDs, SP-08-L1). This will help ensure long-term availability to the specific parts, which is especially important for this custom set-up. All suppliers and product numbers were added to Supp. Table 2 to help with longer-term availability.

Addressed.

24. Line 616 and elsewhere: Italics for bacterial species names.

We have made sure italics are used for all bacterial species names

Formatting issues remain in references section.

Italics has been added to reference 13, 22, 23, 28, 43, and 44.

See citation #9.

25. Line 626 Fig S3 labels: Instead of (or in addition to) "green, blue, and red", it would be useful to label how the colors correspond to the gene knockouts.

Thank you for the suggestion, these labels have been changed.
Addressed.

26. Fig S5 legend. Much of this content would be more appropriately placed into the Background or Discussion sections:

a. "BstA, which is encoded by the temperate phage BTP1, acts as a defense system against phages invading the bacterial host in which it has integrated. However, when BTP1 excises from the chromosome to lyse its host cell, it has an anti-defense region that allows for its life cycle to continue unimpeded. For this reason, BTP1 is expected to be able to lyse a cell with BstA on the chromosome while BTP1 Δ *bstA* is not [29]. The blue strain contains a BstA sequence with an early stop codon while the red contains a functioning BstA defense system. As is anticipated, BTP1 makes a clear plaque while BTP1 Δ *bstA* has red fluorescent signal within the plaque."

We agree that this information is not required in the figure legend and have removed a lot of this information from the figure legend of Supp. Figure 6. More details on the defense system can be found in the corresponding citation.

Addressed.

27. You might consider naming or systematically numbering your newly-discovered phages. That would help future work on these isolates reference back to this manuscript. As this manuscript focuses on the method rather than the biology of these phages, we have elected not to name or fully characterize the phages in this paper.

I suppose that's reasonable, although it might reduce how much your work is cited.

28. References: Typos and formatting throughout.

We have fixed all typos we could find in the references and updated them to ASM style format.

Mostly addressed, although see #24.

Italics has been added to reference 13, 22, 23, 28, 43, and 44.

29. To the students: It is great to read a grad student's first, first-author manuscript and to see that undergrads are co-authors as well. We reviewers know you put in many long days, generated a whole lot of data that are not in this manuscript, and conducted countless protocol development assays that aren't even in the supplement. Those are the days and the data where a lot of science happens. You are doing excellent work. I look forward to future characterization of these new phages and seeing where Phage DisCo goes next.

Thank you, it's been a long journey to get here.

:)